# Prolonged cell cycle arrest in response to DNA damage in yeast requires the maintenance of DNA damage signaling and the spindle assembly checkpoint

**Felix Y Zhou[1†], David P Waterman[1†], Marissa Ashton[1], Suhaily Caban-Penix[1], Gonen Memisoglu[1,2], Vinay V Eapen[1‡], James E Haber[1]\***

[1]Department of Biology and Rosenstiel Basic Medical Sciences Research Center, Brandeis University, Waltham, United States; [2]Department of Molecular Genetics & Cell Biology, University of Chicago, Chicago, United States

**\*For correspondence:**
haber@brandeis.edu

[†]These authors contributed equally to this work

**Present address:** [‡]Flagship Pioneering, Cambridge, United States

## eLife Assessment

This is an **important** study on the damage-induced checkpoint maintenance and termination in budding yeast that provides novel and **convincing** evidence for a role of the spindle assembly checkpoint and mitotic exit network in halting the cell cycle after prolonged arrest in response to irreparable DNA double-strand breaks (DSBs). The study identifies particular components from these checkpoints that are specifically required for the establishment and/or the maintenance of a cell cycle block triggered by such DSBs. The authors propose an interesting model for how these different checkpoints intersect and crosstalk for timely resumption of cell cycling even without repairing DNA damage that has been revised by addressing the bulk of the reviewers' comments to the first version of the manuscript.

**Abstract** Cells evoke the DNA damage checkpoint (DDC) to inhibit mitosis in the presence of DNA double-strand breaks (DSBs) to allow more time for DNA repair. In budding yeast, a single irreparable DSB is sufficient to activate the DDC and induce cell cycle arrest prior to anaphase for about 12–15 hr, after which cells 'adapt' to the damage by extinguishing the DDC and resuming the cell cycle. While activation of the DNA damage-dependent cell cycle arrest is well understood, how it is maintained remains unclear. To address this, we conditionally depleted key DDC proteins after the DDC was fully activated and monitored changes in the maintenance of cell cycle arrest. Degradation of Ddc2[ATRIP], Rad9, Rad24, or Rad53[CHK2] results in premature resumption of the cell cycle, indicating that these DDC factors are required both to establish and maintain the arrest. Dun1 is required for the establishment, but not the maintenance, of arrest, whereas Chk1 is required for prolonged maintenance but not for initial establishment of the mitotic arrest. When the cells are challenged with two persistent DSBs, they remain permanently arrested. This permanent arrest is initially dependent on the continuous presence of Ddc2, Rad9, and Rad53; however, after 15 hr these proteins become dispensable. Instead, the continued mitotic arrest is sustained by spindle assembly checkpoint (SAC) proteins Mad1, Mad2, and Bub2 but not by Bub2's binding partner Bfa1. These data suggest that prolonged cell cycle arrest in response to 2 DSBs is achieved by a handoff from the DDC to specific components of the SAC. Furthermore, the establishment and maintenance of DNA damage-induced cell cycle arrest require overlapping but different sets of factors.

## Introduction

DNA double-strand breaks (DSBs) are one of the most deleterious forms of DNA damage (*Mehta and Haber, 2014*). In response to DSBs, cells evoke the DNA damage checkpoint (DDC) to halt the metaphase to anaphase transition (known as the $G_2$/M checkpoint). Activation of DDC gives cells an extended opportunity to repair DSBs and, therefore, prevents the inheritance of broken chromosomes, which can lead to aneuploidy, chromosome aberrations, and genome instability (*Waterman et al., 2020*).

In budding yeast, a single irreparable DSB is sufficient to trigger the DDC through the activation of Mec1, a PI3K-like kinase and homolog of the mammalian ATR (*Mantiero et al., 2007*; *Pellicioli et al., 2001*). Mec1 activation depends on 5' to 3' resection of the DSB ends that exposes single-stranded DNA (ssDNA), which is rapidly coated by the ssDNA binding protein, RPA (reviewed by *Maréchal and Zou, 2015*). As resection proceeds, the PCNA-related 9-1-1 clamp, made up of Ddc1, Rad17, and Mec3, is loaded at the resected ss/dsDNA junction by the Rad24-Rfc2-5 clamp loader (*Ellison and Stillman, 2003*; *Majka et al., 2006*). Mec1 is recruited to DSB sites via its obligate binding partner, Ddc2$^{ATRIP}$ interacting with RPA-bound ssDNA (*Dubrana et al., 2007*; *Zou and Elledge, 2003*). Following its localization to DSBs, Mec1's kinase activity is stimulated by Dbp11, Dna2, and the Ddc1 subunit of the 9-1-1 clamp (*Navadgi-Patil and Burgers, 2009*; *Melo et al., 2001*). Impairing Mec1's kinase activity by the PI3K-like kinase inhibitor caffeine, by using temperature-sensitive Mec1 mutants or by degradation of Mec1's binding partner Ddc2, rapidly extinguishes checkpoint signaling (*Pellicioli et al., 2001*; *Tsabar et al., 2015*; *Vaze et al., 2002*), illustrating that continual Mec1 activity is needed to activate and sustain DDC. In contrast to Mec1, yeast's other PI3K-like kinase, Tel1, the homolog of mammalian ATM, is dispensable for DDC activation and maintenance, as *TEL1* deletion only shortens damage-induced cell cycle arrest by a few hours (*Dubrana et al., 2007*).

Following the induction of a DSB, numerous proteins are phosphorylated either directly by Mec1 or by the downstream effector kinases Rad53 and Chk1 (human CHK2 and CHK1), which are themselves Mec1 substrates (*Lanz et al., 2019*; *Smolka et al., 2007*). Mec1 and Tel1 substrates also include histone H2A-S129, called γ-H2AX, which spreads on both sides of the DSB via two distinct mechanisms (*Rogakou et al., 1998*; *Shroff et al., 2004*). γ-H2AX then recruits the scaffold protein Rad9, which brings the effector kinase Rad53 in close proximity to Mec1 for activation (*Durocher et al., 1999*; *Emili, 1998*; *Schwartz et al., 2002*). Activated Rad53 then amplifies the DDC signal through autophosphorylation in trans, also stimulating the transcription regulator Dun1 kinase, while restraining the degradation of Pds1 (securin) to inhibit mitosis (*Chen et al., 2007*; *Fiorani et al., 2008*; *Pellicioli et al., 1999*; *Usui and Petrini, 2007*; *Yam et al., 2020*).

In addition to the DDC, unattached kinetochores can induce cell cycle arrest through the activation of spindle assembly checkpoint (SAC) (reviewed by *Musacchio, 2015*). Several studies have suggested a crosstalk between the SAC and the DDC. For example, deletion of SAC components *MAD1* or *MAD2* shortens the cell cycle arrest in response to DNA-damaging agents in the absence of DDC genes *RAD9* and *RAD24* (*Garber and Rine, 2002*; *Kim and Burke, 2008*). Furthermore, *MAD1*, *MAD2*, or *BUB1* mutants arrest for less time than wild-type cells following the induction of a single persistent DSB (*Dotiwala et al., 2010*). In mouse oocytes, inhibition of the SAC overrides the activation of DDC-mediated metaphase arrest during the first meiotic division (*Marangos et al., 2015*). The mitotic exit network (MEN) is another signaling cascade activated during anaphase to promote cell cycle re-entry *Matellán and Monje-Casas, 2020*; therefore, defects in MEN lead to mitotic arrest in late anaphase (*Geymonat et al., 2002*; *Bardin et al., 2000*; *Shirayama et al., 1994*). In addition to SAC, MEN also communicates with the DDC in response to DNA damage. For instance, a key regulator of MEN, the heterodimer Bub2/Bfa1 complex, is modified in a Rad53 and Dun1-dependent manner following damage (*Hu et al., 2001*). Supporting the idea of crosstalk between MEN and DDC, our lab had shown that deletion of *BUB2* shortened the duration of the arrest in response to a single unrepaired DSB (*Dotiwala et al., 2010*).

Here, we present new mechanistic insights into the *maintenance* of the cell cycle arrest following DNA damage by employing the auxin-inducible degron (AID) strategy to conditionally deplete DDC and SAC proteins. An advantage of the AID system, compared to null or temperature-sensitive mutants, is that AID-tagged proteins retain wild-type function until the addition of the plant hormone auxin (indole-3-acetic acid [IAA]), which triggers rapid degradation of AID-tagged proteins in the presence of the TIR1 E3 ubiquitin ligase (*Morawska and Ulrich, 2013*; *Nishimura et al., 2009*). To

investigate how conditional depletion of DDC or SAC proteins alters the maintenance of cell cycle arrest, we engineered a yeast strain that permanently arrests due to the presence of two persistent DNA breaks. We find that the maintenance of the cell cycle arrest requires constant presence of some, but not all, checkpoint activation proteins. Surprisingly, we find that the DDC proteins that are essential to induce the cell cycle arrest and sustain it at the early stages of the arrest become dispensable nearly 15 hr after DNA damage induction. Conversely, SAC proteins are dispensable for the establishment and the initial steps of the cell cycle arrest but become essential at later stages of the DNA damage-dependent cell cycle arrest. Based on these findings, we posit that prolonged cell cycle arrest in response to DNA damage is sustained by both SAC and DDC; however, each checkpoint sustains the arrest at different stages.

## Results

### Measuring DNA damage checkpoint arrest and maintenance

To study the role of DDC initiation proteins in the *maintenance* of the cell cycle arrest, we utilized the well-characterized strain JKM179 (*Lee et al., 1998*), in which the site-specific HO endonuclease is expressed from a *GAL1-10* promoter (*GAL-HO*) to induce a single DSB within the *MAT* locus on chromosome III (referred to as the 1-DSB strain). In this 1-DSB strain, we inserted an additional HO cleavage site 52 kb from the centromere on chromosome IV to induce another DSB (referred to as the 2-DSB strain) (*Kim et al., 2007*; *Lee et al., 1998*). At both loci, HO-mediated cleavage after galactose induction is nearly complete within 30–45 min (*Lee et al., 2014*). In both strains, we also deleted the *HML* and *HMR* donors to prevent repair by homologous recombination. With continuous *HO* expression, nonhomologous end-joining occurs in only 0.2% of these cells (*Moore and Haber, 1996*), thus both DSBs are essentially irreparable.

Following the induction of 2 DSBs, we monitored cell cycle arrest in four ways: (1) with an adaptation time-course assay where we micromanipulated individual $G_1$ cells on agar plates and scored the percentage of cells that are able to re-enter mitosis (*Lee et al., 1998*), (2) by monitoring the percentage of $G_2$/M-arrested cells in liquid culture based on cell morphology (*Figure 1A*), (3) by monitoring nuclear division by DAPI staining of nuclei, and (4) by assaying Rad53 phosphorylation by western blot (*Pellicioli et al., 2001*).

In both 1-DSB and 2-DSB strains, 4 hr after the induction of DNA damage, >90% of cells arrested at $G_2$/M as determined by an adaptation assay (*Figure 1B and E*) and DAPI staining (*Figure 1D and G*). In agreement, western blot analysis showed that Rad53 was hyperphosphorylated (*Figure 1C and F*), demonstrating that DDC was fully activated in both these strains after DNA damage. By 12–15 hr after the induction of a single persistent DSB, most 1-DSB cells adapted; that is, they escaped the $G_2$/M arrest and re-entered mitosis (*Figure 1B*). The timing of Rad53 dephosphorylation following the induction of a single irreparable DSB correlated with the timing of adaptation and escape from the $G_2$/M arrest (*Figure 1C*), as previously shown (*Pellicioli et al., 2001*) In contrast, in the 2-DSB strain, over 90% of cells remained permanently arrested in $G_2$/M with persistently hyper-phosphorylated Rad53 throughout the 24 hr time course (*Figure 1E–G*). We leveraged this permanent cell cycle arrest observed in the 2-DSB strain to study how the DNA damage-induced cell cycle arrest is maintained once it had been established.

### Analysis of checkpoint factors required for checkpoint maintenance

We used the AID system (*Nishimura et al., 2009*) to conditionally deplete DDC proteins after $G_2$/M arrest had been established to study the maintenance of cell cycle arrest. To this end, we appended an AID tag with nine copies of the c-MYC epitope tag to the C-terminus of Ddc2, Rad9, Rad24, and Rad53, which are all components of the Mec1 signaling cascade (*Memisoglu et al., 2019*; *Sweeney et al., 2005*; *de la Torre-Ruiz et al., 1998*). Hereafter, all AID-tagged proteins will be designated simply as -AID, for example, Ddc2-9xMyc-AID as Ddc2-AID.

AID tagging of DDC proteins did not alter the establishment of $G_2$/M arrest; however, *RAD9-AID*, *RAD24-AID*, and *RAD53-AID* strains are hypomorphic and adapted 24 hr after DNA damage in the absence of IAA, while 2-DSB wild-type counterpart cells remained fully arrested (*Figure 2—figure supplement 1A*). This premature escape from the cell cycle arrest was dependent on the presence of TIR1 (*Figure 2—figure supplement 1B*) and is likely due to low levels of IAA as a natural intermediate

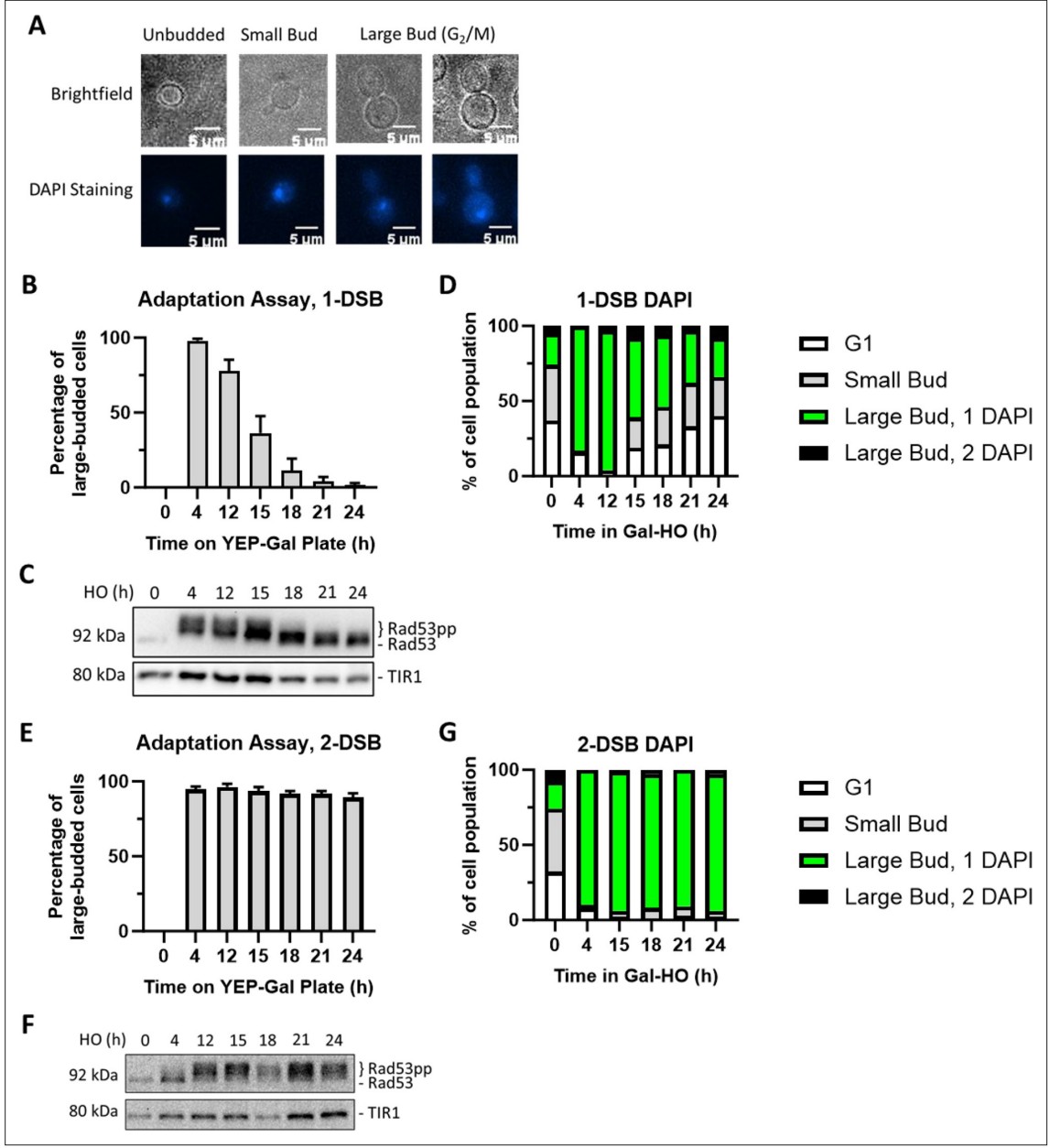

**Figure 1.** Measuring checkpoint arrest in 1-DSB and 2-DSB strains. (**A**) Morphological categories of budding yeast cells using brightfield microscopy and DAPI staining were used to determine G$_2$/M arrest. Cells that arrest at G2/M shift toward a large bud state. G$_2$/M-arrested cells that progress into anaphase. (**B**) Adaptation assay with 1-DSB strain on a YEP-Gal plate. G$_2$/M arrest was determined based on cell morphology as shown in (**A**). Data are shown from three independent experiments, error bars represent standard error of the mean (SEM). (**C**) Profile of DAPI-stained cells in a 1-DSB strain after DNA damage induction in liquid culture. Cells were grouped based on cell morphology and DAPI staining profiles, as explained below the graphs. (**D**) Rad53 phosphorylation kinetics in 1-DSB strain by western blotting. Samples collected after the induction of DNA damage during the time-course experiment and blotted with α-Rad53 to monitor DDC signaling. α-Rad53 can both detect unphosphorylated and hyperphosphorylated Rad53 species. TIR1-Myc was detected with α-Myc and serves as a loading control. (**E**) Same as (**B**) for a 2-DSB strain. (**F**) Same as (**C**) with a 2-DSB strain. (**G**) Same as (**D**) with a 2-DSB strain.

The online version of this article includes the following source data for figure 1:

**Source data 1.** Original membranes corresponding to *Figure 1C*.

**Source data 2.** Original files corresponding to *Figure 1C*.

**Source data 3.** Original membranes corresponding to *Figure 1F*.

**Source data 4.** Original files corresponding to *Figure 1F*.

in amino acid biosynthesis (*Rao et al., 2010*). IAA treatment prior to the induction of 2 DSBs in a control strain, which does not contain any AID-tagged proteins, did not alter the prolonged cell cycle arrest following DNA damage (*Figure 2—figure supplement 2A*), illustrating that IAA treatment by itself does not alter response to DNA damage. However, rapid degradation of Ddc2-AID, Rad9-AID, Rad24-AID, and Rad53-AID by IAA treatment 2 hr prior to the induction of DSBs largely prevented cell cycle arrest as well as DDC signaling, evident from the absence of detectible Rad53 hyperphos-phorylation (*Figure 2—figure supplement 2B–E*). These results underline the importance of Ddc2, Rad9, Rad24, and Rad53 in initiating DDC signaling and cell cycle arrest in response to DNA damage, agreeing with previous reports (*Pellicioli et al., 2001*; *Paciotti et al., 2000*; *Emili, 1998*; *de la Torre-Ruiz et al., 1998*).

To test whether the DDC proteins are required for the *maintenance* of the cell cycle arrest following DNA damage, we employed the AID-tagged strains with 2 DSBs and depleted the AID proteins 4 hr *after* inducing DSBs. In the absence of IAA, *DDC2-AID*, *RAD9-AID*, or *RAD24-AID* strains all activated the DDC signaling 4 hr after DSB induction, with 89–99% of cells arrested in $G_2$/M (*Figure 2A–C*), demonstrating that AID tagging of these proteins did not impair their function. Within 1 hr after IAA treatment, Ddc2-AID, Rad9-AID, or Rad24-AID were all rapidly depleted, which caused a gradual Rad53 dephosphorylation, as detected by western blotting (*Figure 2A–C*). Moreover, agreeing with the loss of Rad53 phosphorylation, IAA treatment of DDC-AID strains triggered release from $G_2$/M arrest, while the untreated control cells remained fully arrested. DAPI staining of *DDC2-AID* cells after IAA treatment revealed the accumulation of large-budded cells with two distinct DAPI signals, indicating that cells started to progress into anaphase following Ddc2 depletion (*Figure 2—figure supplement 3A*). These findings illustrate that the upstream DDC factors Ddc2, Rad9, and Rad24 are essential for initiating and sustaining the cell cycle arrest in response to DNA damage.

Compared to *DDC2-AID*, *RAD9-AID*, or *RAD24-AID*, we found that *RAD53-AID* cells maintained $G_2$/M arrest for an additional 4 hr after complete depletion of Rad53 (*Figure 2D*). In contrast to Ddc2-AID depletion, cell cycle analysis by DAPI staining showed that Rad53 depletion led to a more gradual transition into late anaphase (*Figure 2—figure supplement 3B*). This delay after the conditional depletion of Rad53 could be due to continued signaling from downstream targets activated by Rad53 kinase, such as Dun1, or from other targets of the Mec1 kinase, downstream of Ddc2, Rad9, and Rad24.

## Chk1 sustains checkpoint signaling in the absence of Ddc2, Rad9, Rad24, or Rad53

Rad53 and Chk1 kinases both contribute to the maintenance of the cell cycle arrest after DNA damage (*Dotiwala et al., 2007*; *Pellicioli et al., 2001*). Agreeing with previously published reports (*Sanchez et al., 1999*), we find that Chk1 is involved in maintaining the permanent arrest following the induction of 2 DSBs. Deletion of *CHK1* did not impair the induction of cell cycle arrest (*Figure 3A*); however, it inhibited the permanent cell cycle arrest as >95% of *chk1Δ* cells adapted by 24 hr (*Figure 3B*). We then asked whether the delay in cell cycle re-entry observed when Rad53 was degraded was due to Chk1's independent role in maintaining arrest. To test this, we induced 2 DSBs in *RAD53-AID chk1Δ* cells for 4 hr and then added IAA to deplete Rad53. Compared to the depletion of Rad53-AID alone (*Figure 2D*), the depletion of Rad53-AID in the absence of *CHK1* led to a significant decrease in the number of $G_2$/M-arrested cells within 1 hr of IAA treatment (*Figure 3C*). These results suggest that Chk1 functions in conjunction with Rad53 to sustain cell cycle arrest in response to DNA damage.

To explore further how Chk1 signaling contributes to the maintenance of DDC-dependent cell cycle arrest, we depleted DDC factors Ddc2-AID, Rad9-AID, or Rad24-AID in *chk1Δ* cells 4 hr after the induction of DSBs. Depletion of these upstream factors in the absence of *CHK1* led to a more rapid release from the cell cycle arrest (*Figure 3D–F*) compared to the depletion of DDC factors alone (*Figure 2A–C*). Collectively, these findings demonstrate that Chk1 plays a key role in maintaining cell cycle arrest in response to DNA damage.

Tel1 is thought to play a minor role in response to DSBs as the establishment of DSB-induced cell cycle arrest normally depends entirely on Mec1 (*Dotiwala et al., 2010*). However, previous studies have shown that, in addition to Mec1, Tel1 can also target Chk1 for phosphorylation (*Limbo et al., 2011*; *Sanchez et al., 1999*). Additionally, when the initial 5′ to 3′ end resection of DSB ends is impaired, Tel1 alone can activate the DDC (*Usui and Petrini, 2007*). To study how Tel1 contributes to

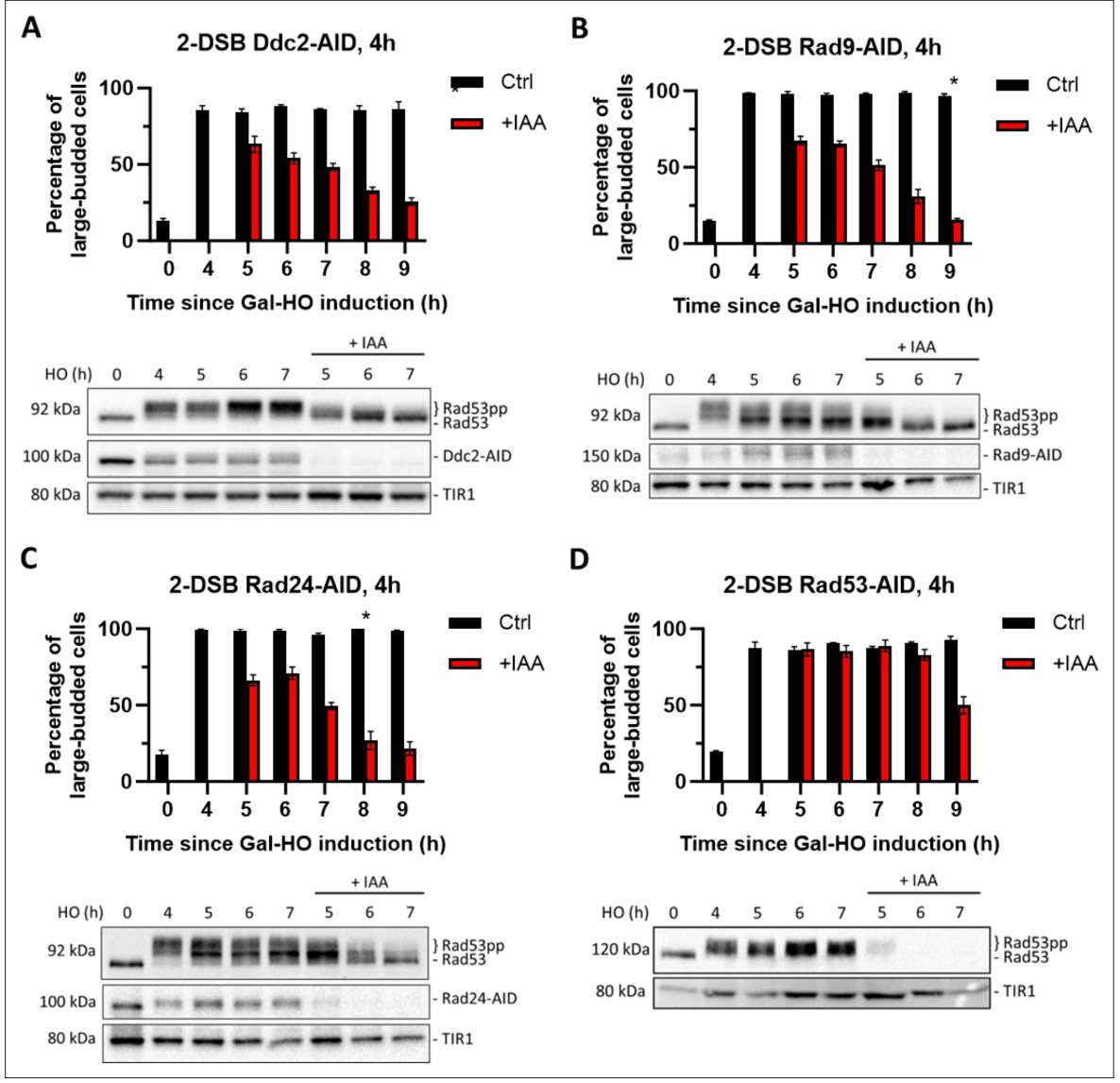

**Figure 2.** Checkpoint maintenance requires Ddc2, Rad9, Rad24, and Rad53 activity. (**A**) Above: percentage of $G_2$/M-arrested cells in a 2-DSB *DDC2-AID* strain after DNA damage induction in a liquid culture. Cultures were split 4 hr after galactose treatment to induce DNA damage by GAL::HO and treated either with auxin (+IAA) (1 mM) or with ethanol (Ctrl). Data are shown from three independent experiments, with error bars representing standard error of the mean (SEM). The asterisk marks the timepoint when the percentage of large-budded $G_2$/M cells returned to pre-damage levels. Below: western blots ran with samples collected at various timepoints during the same time course, probed with α-Rad53, to determine DDC status, and α-Myc, to determine Ddc2-AID-Myc protein abundance and TIR1-Myc as a loading control. (**B**) Same as (**A**) for 2-DSB *RAD9-AID*. (**C**) Same as (**A**) for 2-DSB *RAD24-AID*. (**D**) Same as (**A**) for 2-DSB *RAD53-AID*.

The online version of this article includes the following source data and figure supplement(s) for figure 2:

**Source data 1.** Original membranes corresponding to *Figure 2A*.

**Source data 2.** Original files corresponding to *Figure 2A*.

**Source data 3.** Original membranes corresponding to *Figure 2B*.

**Source data 4.** Original files corresponding to *Figure 2B*.

**Source data 5.** Original membranes corresponding to *Figure 2C*.

**Source data 6.** Original files corresponding to *Figure 2C*.

**Source data 7.** Original membranes corresponding to *Figure 2D*.

**Source data 8.** Original files corresponding to *Figure 2D*.

**Figure supplement 1.** Adaptation assay of AID-tagged checkpoint activation proteins.

*Figure 2 continued on next page*

*Figure 2 continued*

**Figure supplement 2.** AID-tagged checkpoint proteins readily degrade with auxin.

**Figure supplement 2—source data 1.** Original membranes corresponding to *Figure 2—figure supplement 2A*.

**Figure supplement 2—source data 2.** Original files corresponding to *Figure 2—figure supplement 2A*.

**Figure supplement 2—source data 3.** Original membranes corresponding to *Figure 2—figure supplement 2B*.

**Figure supplement 2—source data 4.** Original files corresponding to *Figure 2—figure supplement 2B*.

**Figure supplement 2—source data 5.** Original membranes corresponding to *Figure 2—figure supplement 2C*.

**Figure supplement 2—source data 6.** Original files corresponding to *Figure 2—figure supplement 2C*.

**Figure supplement 2—source data 7.** Original membranes corresponding to *Figure 2—figure supplement 2D*.

**Figure supplement 2—source data 8.** Original files corresponding to *Figure 2—figure supplement 2D*.

**Figure supplement 2—source data 9.** Original membranes corresponding to *Figure 2—figure supplement 2E*.

**Figure supplement 2—source data 10.** Original files corresponding to *Figure 2—figure supplement 2E*.

**Figure supplement 3.** Cell cycle profile as determined by budding and DAPI staining in Ddc2-AID and Rad53-AID mutants ±IAA 4 hr after galactose.

the maintenance of the permanent cell cycle arrest, we deleted *TEL1* in 2-DSB strain. Unlike *chk1Δ* with 2 DSBs, a *TEL1* deletion did not affect either the establishment of the DDC or the maintenance of checkpoint arrest up to 24 hr (*Figure 3—figure supplement 1*), agreeing with previously published results (*Dubrana et al., 2007*). Taken together, these data illustrate that DNA damage-dependent cell cycle arrest is initiated by Mec1 branch of the DDC via Ddc2, Rad9 and Rad24, and the cell cycle arrest is largely sustained by the downstream kinases Rad53 and Chk1, with minor contributions from other downstream targets of DDC.

## Dun1 is required for the initiation but not for the maintenance of cell cycle arrest

Our findings show that a small number of cells remain arrested in the absence of *CHK1* when Rad53 is depleted. We posited that Rad53 could modulate the expression of other DDC factors, which, in turn, sustain the cell cycle arrest in the absence of Rad53 and Chk1. One candidate protein is Dun1, a Rad53-activated protein kinase that regulates transcription in response to DNA damage (*Chen et al., 2007*; *Yam et al., 2020*; *Zhou and Elledge, 1993*). Deleting *DUN1* significantly impaired checkpoint activation: compared to the wild-type control strain, only 60% of *DUN1* cells arrested in $G_2$/M 4 hr after the DSB induction, and only 25% remained in $G_2$/M arrest at 7 hr (*Figure 4A*). Additionally, depletion of Dun1-AID 4 hr after damage induction did not cause a significant change in the proportion of $G_2$/M arrested in these otherwise wild-type cells, nor did it affect Rad53 phosphorylation (*Figure 4B*); however, deleting *CHK1* triggered an exit from checkpoint arrest following the depletion of Dun1-AID 4 hr after DSB induction (*Figure 4C*), further demonstrating the role of Chk1 in checkpoint maintenance. These results suggest that Dun1 is required for the initiation of DDC and concomitant cell cycle arrest but is dispensable for maintenance.

## Ddc2 and Rad53's role in maintaining arrest become dispensable in extended $G_2$/M arrest

Previously, we reported that Ddc2 protein abundance initially increases over time as after the induction of a single DNA break, which is followed by near-complete depletion of Ddc2 around the time that cells adapt (*Memisoglu et al., 2019*). Given that Ddc2 overexpression leads to permanent cell cycle arrest following DNA damage (*Clerici et al., 2001*), we concluded that Ddc2 abundance is intimately tied to the duration of the arrest. Here, we asked whether the presence of 2 DSBs instead of a single DSB would lead to an increase in Ddc2 protein abundance and, therefore, the permanent cell cycle arrest. We examined the levels of Ddc2 in both 1- and 2-DSB strains following DNA damage but did not detect a difference in the changes of abundance of Ddc2 protein (*Figure 5—figure supplement 1*), even though cells adapt to 1 DSBs and remained terminally arrested after 2 DSBs (*Figure 1A–D*).

If Ddc2 activity is essential to maintain the cell cycle arrest in the 2-DSB strain, then degradation of Ddc2-AID around the time wild-type cells adapt to a single DNA break should interrupt the permanent cell cycle arrest and trigger cell cycle re-entry. We find that complete depletion of Ddc2-AID 15 hr

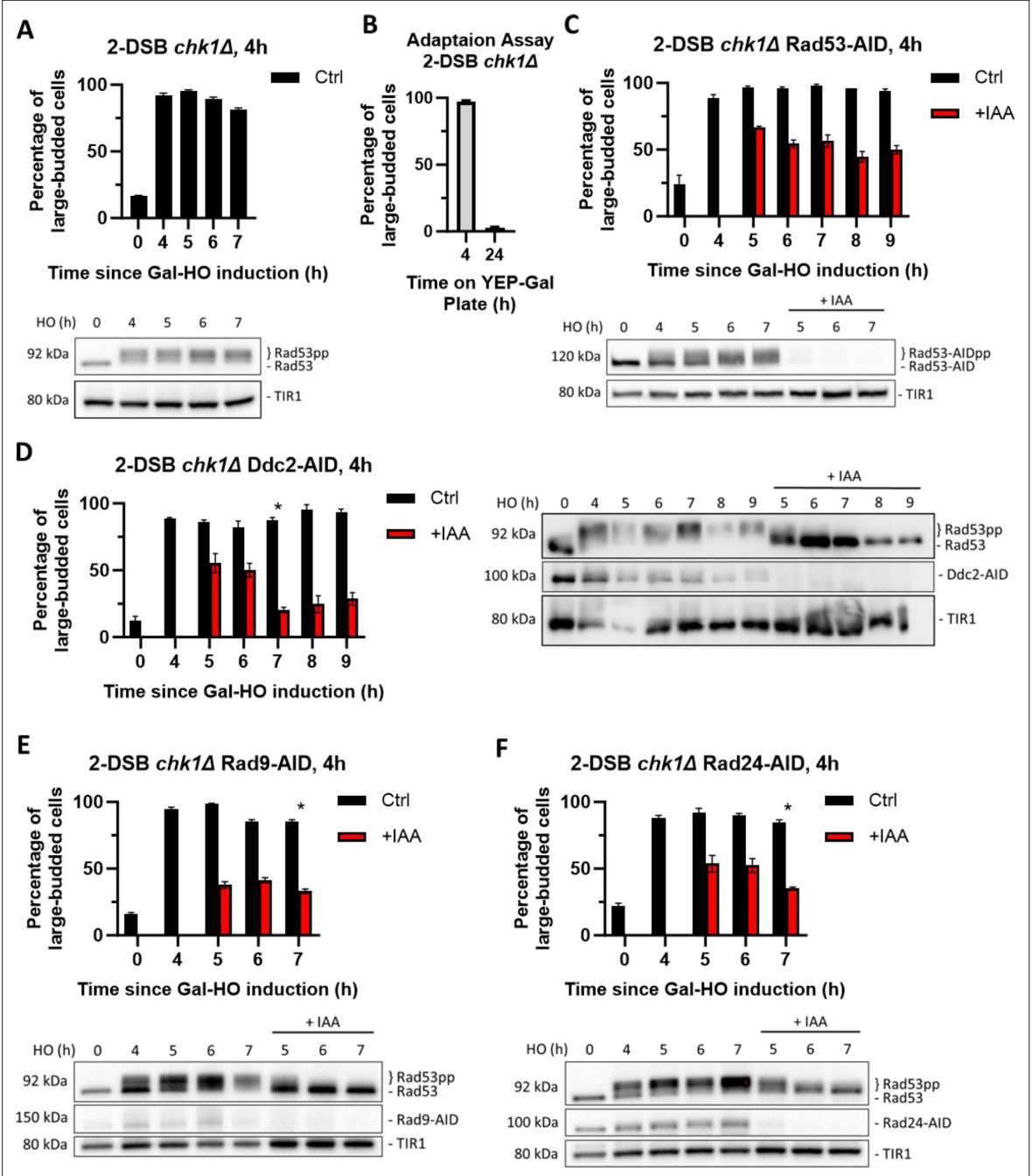

**Figure 3.** Chk1 is dispensable for activation of the cell cycle arrest, but essential for its maintenance. (**A**) Percentage of G$_2$/M cells in a 2-DSB *chk1Δ* strain following DNA damage. Data are shown from three independent experiments, with error bars representing the standard error of the mean (SEM). Western blot probed with α-Rad53 to determine the status of DDC and α-Myc to determine TIR1-Myc protein abundance. (**B**) Adaptation assay with 2-DSB *chk1Δ* strain. (**C**) Percentage of G$_2$/M-arrested cells a 2-DSB *chk1Δ RAD53-AID* strain after DNA damage. Cultures were split 4 hr after DSB induction and treated with 1 mM auxin (+IAA) or with ethanol (Ctrl). Data are shown from three independent experiments, with error bars representing the standard error of the mean (SEM). Western blot probed with α-Myc for Rad53-AID and TIR1-Myc as a loading control. (**D**) Same as (**C**) for 2-DSB *chk1Δ DDC2-AID*. Western blot probed with α-Rad53 and α-Myc. α-Rad53 shows both an unphosphorylated protein and multiple phosphorylated species. α-Myc shows Ddc2-AID degradation and TIR1-Myc as a loading control. The asterisk shows when the percentage of large-budded cells returned to pre-damage levels. (**E**) Same as (**D**) for 2-DSB *chk1Δ RAD9-AID*. (**F**) Same as (**D**) for 2-DSB *chk1Δ RAD24-AID*.

The online version of this article includes the following source data and figure supplement(s) for figure 3:

**Source data 1.** Original membranes corresponding to *Figure 3A*.

**Source data 2.** Original files corresponding to *Figure 3A*.

*Figure 3 continued on next page*

*Figure 3 continued*

**Source data 3.** Original membranes corresponding to *Figure 3C*.

**Source data 4.** Original files corresponding to *Figure 3C*.

**Source data 5.** Original membranes corresponding to *Figure 3D*.

**Source data 6.** Original files corresponding to *Figure 3D*.

**Source data 7.** Original membranes corresponding to *Figure 3E*.

**Source data 8.** Original files corresponding to *Figure 3E*.

**Source data 9.** Original membranes corresponding to *Figure 3F*.

**Source data 10.** Original files corresponding to *Figure 3F*.

**Figure supplement 1.** Tel1 is not required for DDC activation or Rad53 phosphorylation.

**Figure supplement 1—source data 1.** Original membranes corresponding to *Figure 3—figure supplement 1B*.

**Figure supplement 1—source data 2.** Original files corresponding to *Figure 3—figure supplement 1B*.

after the induction of 2 DSBs leads to diminished Rad53 hyperphosphorylation (*Figure 5A*). However, surprisingly, Ddc2-AID degradation did not alter the percentage of G$_2$/M-arrested cells even 9 hr after Ddc2 depletion (*Figure 5A*). Furthermore, despite the diminished Rad53 phosphorylation, Ddc2-AID cells mostly remained arrested in G$_2$/M after the depletion of Ddc2-AID at 15 hr as illustrated by DAPI staining (*Figure 5B*), in contrast to Ddc2-AID depletion soon after induction of 2 DSBs, which causes cells to rapidly resume mitosis (*Figure 2A*). These results hint that the maintenance of the permanent cell cycle arrest in response to 2 DSBs at later stages could be independent of the DDC signaling.

We then asked whether other DDC factors such as Rad9, Rad24, and Rad53 are dispensable for the prolonged arrest following the induction of 2 DSBs. However, as noted above, AID-tagged DDC activation proteins were unable to maintain this prolonged arrest in a 2-DSB strain even in the absence of IAA, which precludes their use in this analysis. To be able to study the contribution of these DDC factors in prolonged cell cycle arrest, we turned to the AID version 2 (AID2) system (*Yesbolatova et al., 2020*). To this end, we integrated a TIR1-F74G point mutation and used 5-phenyl-IAA (5-Ph-IAA) instead of IAA to lower the basal degradation of AID-tagged proteins. Switching to the AID2 system did not fully restore function to *RAD9-AID2* and *RAD24-AID2* strains as they mostly adapted after 24 hr after the exposure to DNA damage, but 87% of *RAD53-AID2* cells remained in G$_2$/M arrest at 24 hr (*Figure 2—figure supplement 1C*). Using Rad53-AID2, we then asked whether Rad53 is required for extended G$_2$/M arrest in response to 2 DSBs. Rapid depletion of Rad53-AID2 with 5-Ph-IAA 4 hr after DSB induction led cells to escape G$_2$/M arrest, but as with the *RAD53-AID* strain, we detected a 4 hr delay in cell cycle re-entry, confirming our previous results (*Figures 2D and 5C*). However, degradation of Rad53-AID2 15 hr after DSB induction did not prompt cell cycle re-entry even 9 hr after complete depletion of Rad53 (*Figure 5D*, *Table 1*), akin to what we observe following the depletion of Ddc2-AID (*Figure 5A and B*).

Because TIR1-mediated degradation of Rad9-AID even without auxin caused most cells to adapt 24 hr after inducing DNA damage, we asked whether overexpression of *RAD9-AID* could overcome this effect. We added a *TRP1* centromere-containing plasmid copy of *RAD9-AID* with its endogenous promoter (pRAD9-AID) to our 2-DSB *RAD9-AID* strain. Degradation of Rad9-AID by IAA 15 hr after DSB induction did not trigger release of cells from G$_2$/M arrest (*Figure 5E*, *Table 1*). However, unlike degradation of Ddc2-AID or Rad53-AID2 in this same situation, Rad53 remained hyperphosphory-lated up to 9 hr after adding IAA (*Figure 5E*). Therefore, while the DDC proteins Ddc2, Rad9, and Rad53 are required for the maintenance of checkpoint arrest at early stages, surprisingly, they are dispensable for prolonged arrest following induction of 2 DSBs. These results suggest that prolonged cell cycle arrest is maintained by signaling proteins other than the Mec1 branch of the DDC.

## Spindle assembly checkpoint proteins Mad1 and Mad2 are required for prolonged arrest

In addition to the DDC, the SAC maintains genomic integrity by halting mitosis at the metaphase/anaphase transition in response to unattached kinetochores, to ensure accurate chromosome segregation (reviewed by *McAinsh and Kops, 2023*). We have previously shown that inactivation of the SAC by a *MAD1, MAD2,* or *MAD3* deletion shortened the duration of the cell cycle arrest induced

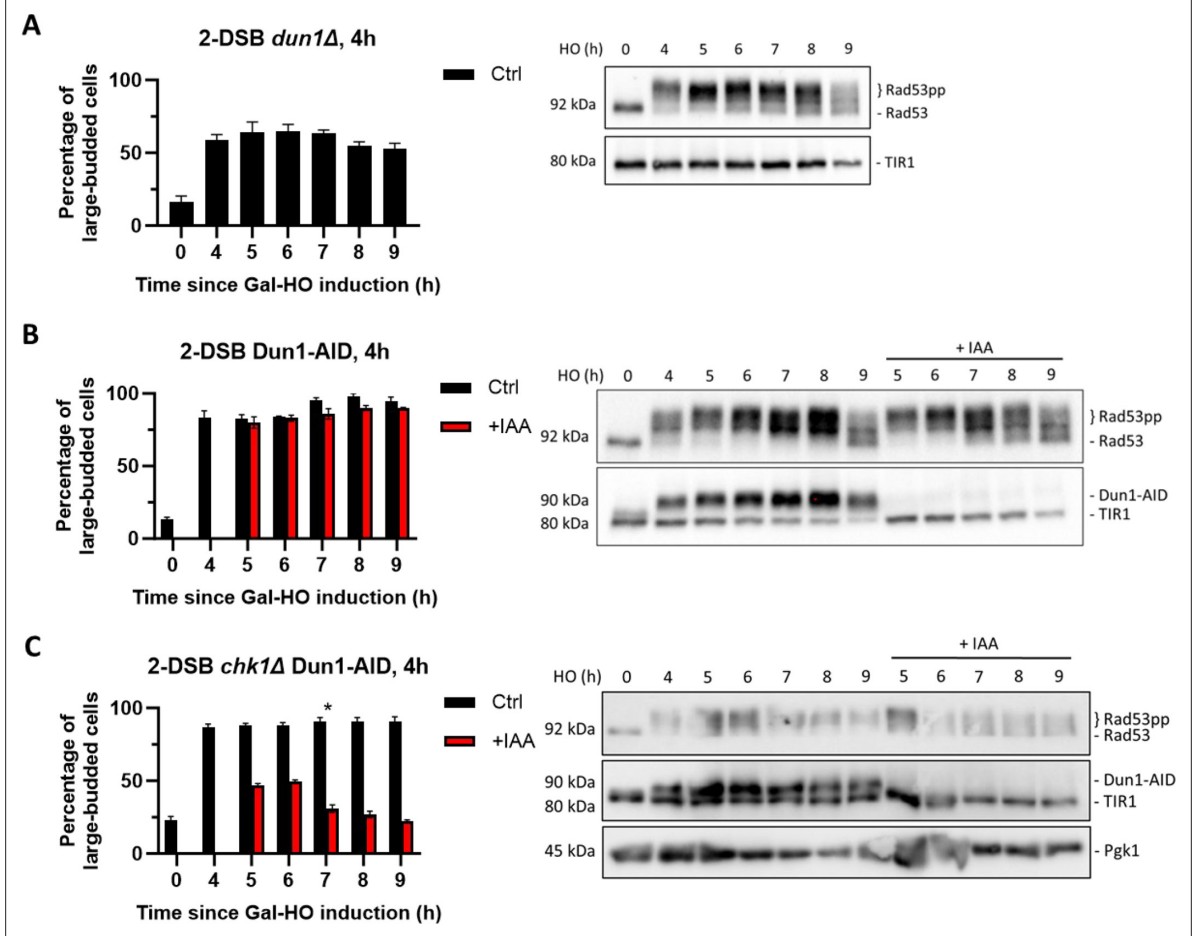

**Figure 4.** Dun1 is not required for checkpoint maintenance. (**A**) Adaptation assay of 50 G$_1$ cells on a YEP-Gal plate with 2-DSB *dun1Δ*. G$_2$/M arrest was determined based on cell morphology as shown in *Figure 1A*. Data is shown from three trials with standard error of the mean (SEM). Western blot probed with α-Rad53 and α-Myc for TIR1-Myc as a loading control. (**B**) Percentage of G$_2$/M-arrested cells for 2-DSB *DUN1-AID* after HO induction. Data are shown from three trials with standard error of the mean (SEM). Cultures were split 4 hr after DSB induction; with auxin (1 mM) (+IAA). Western blot probed with α-Rad53 and α-Myc. α-Rad53 shows both an unphosphorylated protein and multiple phosphorylated species. α-Myc shows Dun1-AID degradation and TIR1-Myc as a loading control. (**C**) Same as (**B**) for 2-DSB *chk1Δ DUN1-AID*. The asterisk marks when the percentage of large-budded cells returned to pre-damage levels.

The online version of this article includes the following source data for figure 4:

**Source data 1.** Original membranes corresponding to *Figure 4A*.

**Source data 2.** Original files corresponding to *Figure 4A*.

**Source data 3.** Original membranes corresponding to *Figure 4B*.

**Source data 4.** Original files corresponding to *Figure 4B*.

**Source data 5.** Original membranes corresponding to *Figure 4C*.

**Source data 6.** Original files corresponding to *Figure 4C*.

by a single DSB (*Dotiwala et al., 2010*). To test whether SAC is involved in enforcing and sustaining permanent cell cycle arrest in response to 2 DSBs, we deleted *MAD2* in the 2-DSB strain. Adaptation assay results illustrate that nearly all *mad2Δ* cells arrested at 4 hr but began to adapt between 12 and 15 hr after DNA damage (*Figure 6—figure supplement 1A*). To explore whether deletion of *MAD2* can antagonize the permanent cell cycle arrest due to hyperactive DDC signaling, we overexpressed Ddc2 in *mad2Δ* 2-DSB cells and assayed mitotic progression. We find that both in 1-DSB and 2-DSB strains, *MAD2* deletion leads to cell cycle re-entry even when Ddc2 is overexpressed (*Figure 6—figure supplement 1B*). Based on these data, we concluded that mitotic inhibition is enforced by the SAC proteins as DDC factors become dispensable 12–15 hr after the induction of damage.

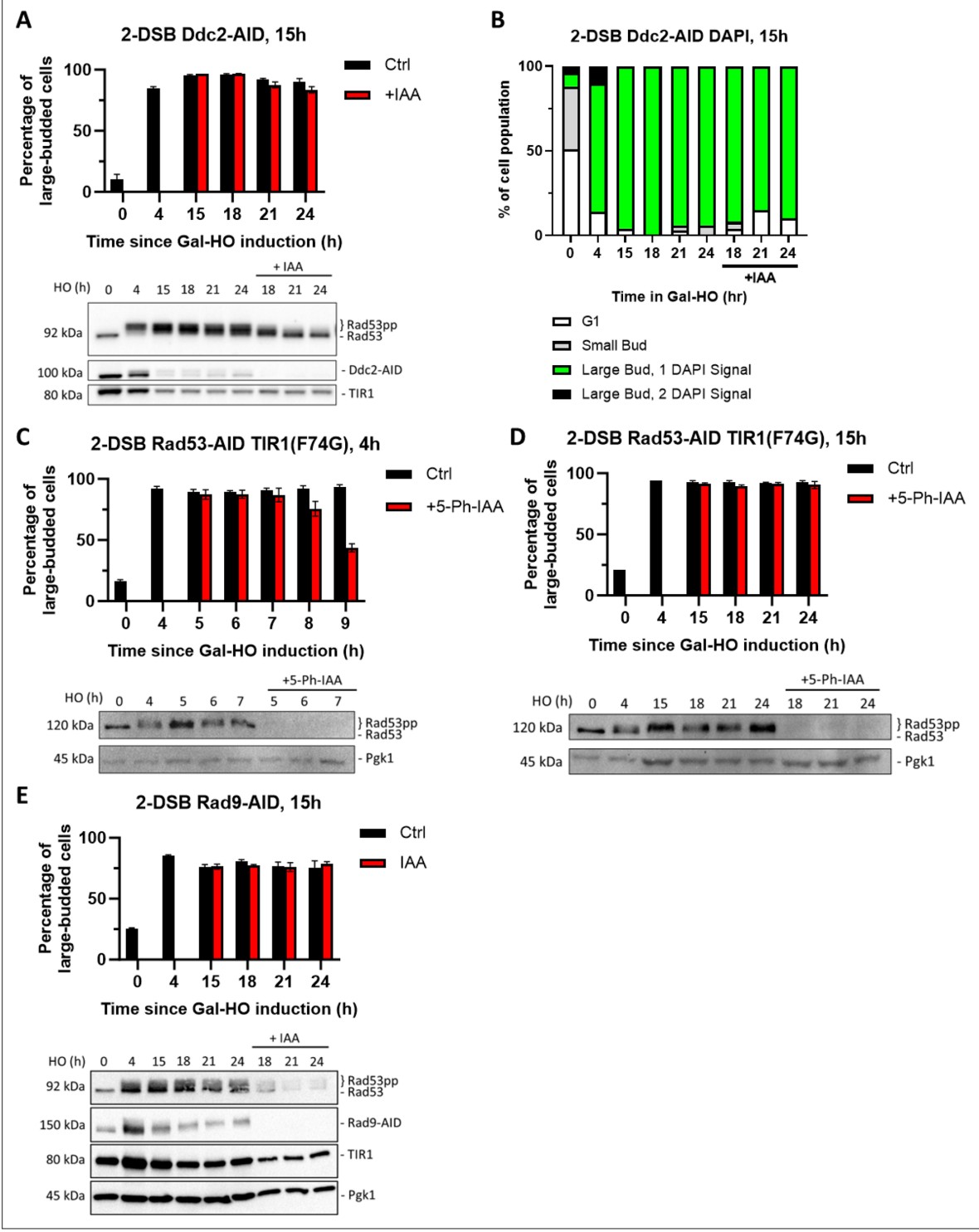

**Figure 5.** Ddc2 and Rad53 are dispensable for >24 hr checkpoint arrest. (**A**) Percentage of G₂/M-arrested cells for 2-DSB *DDC2-AID* after HO induction. Data is shown from three trials with standard error of the mean (SEM). Western blot probed with α-Rad53 and α-Myc. α-Rad53 shows both an unphosphorylated protein and multiple phosphorylated species. α-Myc shows Ddc2-AID degradation and TIR1-Myc as a loading control. (**B**) Profile of DAPI-stained cells in a 2-DSB *DDC2-AID* strain after HO induction. Cells were categorized based on cell morphology and number of DAPI signals. (**C**) Percentage of G₂/M-arrested cells for 2-DSB *RAD53-AID TIR1(F74G)* after HO induction. 5-Ph-IAA was added 4 hr after HO induction. Data is shown from three trials with standard error of the mean (SEM). Western blot probed with α-Rad53, α-Myc, and α-Pgk1. α-Rad53 shows both an unphosphorylated protein and multiple phosphorylated species. α-Myc shows Rad53-AID degradation. α-Pgk1 probed as a loading control. (**D**) Same as (**C**) where 5-Ph-IAA was added 15 hr after HO induction. (**E**) Percentage G₂/M-arrested cells for 2-DSB *RAD9-AID* plus pRad9-AID after HO

*Figure 5 continued on next page*

*Figure 5 continued*

induction. Data shown from three trials with standard error of the mean (SEM). Western blot probed with α-Rad53 and α-Myc. α-Rad53 shows both an unphosphorylated protein and multiple phosphorylated species. α-Myc shows Rad9-AID degradation and TIR1-Myc as a loading control. α-Pgk1 probed as a loading control.

The online version of this article includes the following source data and figure supplement(s) for figure 5:

**Source data 1.** Original membranes corresponding to *Figure 5A*.

**Source data 2.** Original files corresponding to *Figure 5A*.

**Source data 3.** Original membranes corresponding to *Figure 3C*.

**Source data 4.** Original files corresponding to *Figure 3C*.

**Source data 5.** Original membranes corresponding to *Figure 3D*.

**Source data 6.** Original files corresponding to *Figure 3D*.

**Source data 7.** Original membranes corresponding to *Figure 3E*.

**Source data 8.** Original files corresponding to *Figure 3E*.

**Figure supplement 1.** Relative levels of Ddc2 decrease after DSB induction.

**Figure supplement 1—source data 1.** Original membranes corresponding to *Figure 5—figure supplement 1A*.

**Figure supplement 1—source data 2.** Original files corresponding to *Figure 5—figure supplement 1A*.

**Figure supplement 1—source data 3.** Original membranes corresponding to *Figure 5—figure supplement 1B*.

**Figure supplement 1—source data 4.** Original files corresponding to *Figure 5—figure supplement 1B*.

If SAC proteins are only required at later stages of cell cycle arrest when DDC proteins are dispensable, then the depletion of SAC protein Mad2 or its binding partner Mad1 soon after the induction of DNA damage should not affect DDC or cell cycle arrest up to 12–15 hr. In agreement with this idea, after Mad1-AID or Mad2-AID depletion at 4 hr, cells remained arrested up to 15 hr following DSB induction, with persistent Rad53 hyperphosphorylation (*Figure 6—figure supplement 2*). Strikingly, these cells eventually re-entered the cell cycle by 24 hr (20 hr after depletion of Mad1 and Mad2). It is notable that cells resumed cell cycle progression despite persistent Rad53 hyperphosphorylation. This result reinforces our conclusion that Mad1 and Mad2 are not required for the activation and initial maintenance of arrest but are essential for prolonged cell cycle arrest in response to 2 DSBs. These data also suggest that in the absence of Mad1 or Mad2 cells become insensitive to the arrest normally imposed by DDC.

To monitor the effect of SAC proteins Mad1 and Mad2 at late stages of prolonged cell cycle arrest when DDC signaling becomes dispensable, we depleted Mad1-AID or Mad2-AID 15 hr after DSB induction. Following the depletion of Mad1-AID and Mad2-AID, we observed an immediate reduction in the percentage of $G_2$/M-arrested cells (*Figure 6A–C*). DAPI staining confirmed that Mad1-AID or Mad2-AID-deplete cells re-entered the cell cycle, evident from an increase in the G1 cell population as well as an increase in percentage of large-budded cells with 2 DAPI foci (*Figure 6B and D*). Similar to what we observe following Mad1-AID and Mad2-AID depletion 4 hr after damage, this cell cycle re-entry occurred despite persistent Rad53 hyperphosphorylation (*Figure 6A and C*).

To show that the late stages of the permanent cell cycle arrest in response to 2 DSBs is independent of DDC and dependent on SAC, we inactivated DDC by depleting Ddc2-AID together with Mad2-AID. Although the simultaneous depletion of Ddc2-AID and Mad2-AID led to Rad53 dephosphorylation as detected by western blotting, we found no statistically significant difference in the percentage of cells escaping $G_2$/M arrest in response to *DDC2-AID MAD2-AID* double depletion strain compared to Mad2-AID alone (*Figure 6—figure supplement 3* and *Table 1*). Collectively, our findings indicate that DDC initiates and sustains the cell cycle arrest approximately for 15 hr following DNA damage, but after that DDC becomes dispensable and the permanent arrest is sustained by SAC.

## Mitotic exit network proteins Bfa1 and Bub2 have different roles in the DDR

To investigate the possible contribution of the MEN to the maintenance of the extended cell cycle arrest in response to 2 DSBs, we appended AID tags to upstream MEN proteins Bub2 and Bfa1. Degradation of Bub2-AID both 4 hr and 15 hr after induction of 2 DSBs suppressed $G_2$/M arrest

**Table 1.** Comparison of the percentage of large-budded cells back to baseline levels.

| Figure | Strain | Timepoint comparison* | p-Value | Significance | Post hoc test[†] |
|---|---|---|---|---|---|
| *Figure 2A* | *DDC2-AID* | 0 vs 5 + IAA | <0.0001 | **** | Sidak |
| | | 0 vs 6 + IAA | <0.0001 | **** | Sidak |
| | | 0 vs 7 + IAA | <0.0001 | **** | Sidak |
| | | 0 vs 8 + IAA | 0.0009 | *** | Sidak |
| | | 0 vs 9 + IAA | 0.054 | ns | Sidak |
| *Figure 3D* | *DDC2-AID CHK1Δ* | 0 vs 5 + IAA | <0.0001 | **** | Sidak |
| | | 0 vs 6 + IAA | <0.0001 | **** | Sidak |
| | | 0 vs 7 + IAA | 0.10 | ns | Sidak |
| | | 0 vs 8 + IAA | 0.25 | ns | Sidak |
| | | 0 vs 9 + IAA | 0.072 | ns | Sidak |
| *Figure 2B* | *RAD9-AID* | 0 vs 5 + IAA | <0.0001 | **** | Sidak |
| | | 0 vs 6 + IAA | <0.0001 | **** | Sidak |
| | | 0 vs 7 + IAA | <0.0001 | **** | Sidak |
| | | 0 vs 8 + IAA | 0.0055 | ** | Sidak |
| | | 0 vs 9 + IAA | 1.00 | ns | Sidak |
| *Figure 3E* | *RAD9-AID CHK1Δ* | 0 vs 5 + IAA | 0.00050 | *** | Sidak |
| | | 0 vs 6 + IAA | 0.0052 | ** | Sidak |
| | | 0 vs 7 + IAA | 0.21 | ns | Sidak |
| *Figure 2C* | *RAD24-AID* | 0 vs 5 + IAA | <0.0001 | **** | Sidak |
| | | 0 vs 6 + IAA | <0.0001 | **** | Sidak |
| | | 0 vs 7 + IAA | 0.00 | *** | Sidak |
| | | 0 vs 8 + IAA | 0.36 | ns | Sidak |
| | | 0 vs 9 + IAA | 0.90 | ns | Sidak |
| *Figure 3F* | *RAD24-AID CHK1Δ* | 0 vs 5 + IAA | 0.00010 | *** | Sidak |
| | | 0 vs 6 + IAA | 0.00020 | *** | Sidak |
| | | 0 vs 7 + IAA | 0.10 | ns | Sidak |
| *Figure 5D* | *RAD53-AID TIR1(F74G)* | 18 vs 18 + IAA | 0.35 | ns | Sidak |
| | | 21 vs 21 + IAA | 0.96 | ns | Sidak |
| | | 24 vs 24 + IAA | 0.42 | ns | Sidak |
| *Figure 5E* | *RAD9-AID pRAD9-AID* | 18 vs 18 + IAA | 0.80 | ns | Sidak |
| | | 21 vs 21 + IAA | 0.99 | ns | Sidak |
| | | 24 vs 24 + IAA | 0.84 | ns | Sidak |
| *Figure 6—figure supplement 3A* | *DDC2-AID MAD2-AID AND MAD2-AID* | 18 + IAA vs 18 + IAA | 0.95 | ns | Sidak |
| | | 21 + IAA vs 21 + IAA | 0.64 | ns | Sidak |
| | | 24 + IAA vs 24 + IAA | 0.97 | ns | Sidak |

*Timepoints are relative to when galactose was added.
[†]A one-way ANOVA was used to test for significant differences.

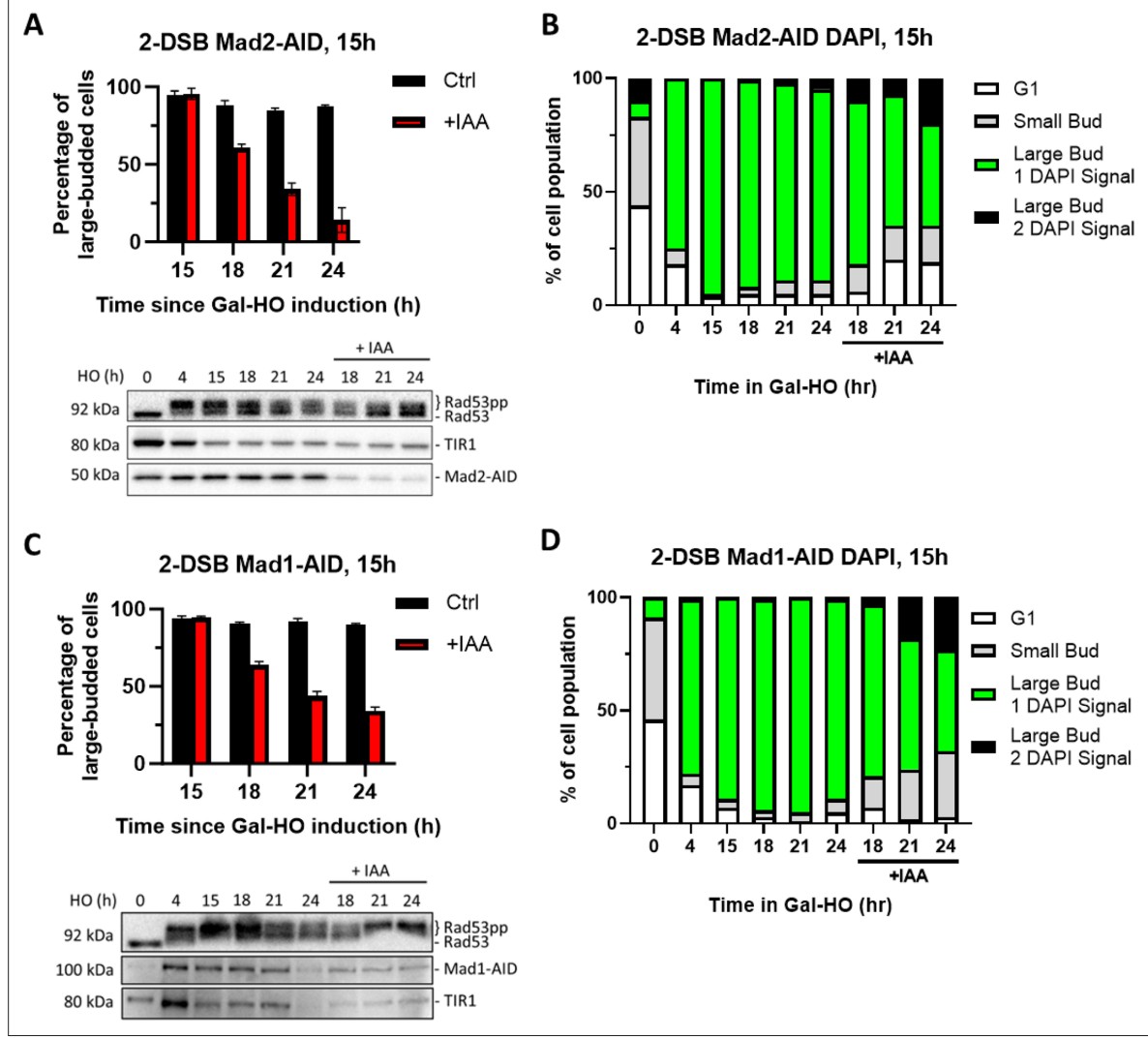

**Figure 6.** Degradation of Mad2 or Mad1 at 15 hr releases cells from checkpoint arrest. (**A**) Percentage of G₂/M-arrested cells for 2-DSB *MAD2-AID* after HO induction. Data is shown from three trials with standard error of the mean (SEM). Western blot probed with α-Rad53 and α-Myc. α-Rad53 shows both an unphosphorylated protein and multiple phosphorylated species. α-Myc shows Mad2-AID degradation and TIR1-Myc as a loading control. (**B**) Profile of DAPI-stained cells in a 2-DSB *MAD2-AID* strain after HO induction. Liquid cultures were split 15 hr after HO induction and treated with either IAA or ethanol. Cells were scored based on cell morphology and number of DAPI signals. (**C**) Same as (**A**) for 2-DSB *MAD1-AID*. (**D**) Same as (**B**) for 2-DSB *MAD1-AID*.

The online version of this article includes the following source data and figure supplement(s) for figure 6:

**Source data 1.** Original membranes corresponding to *Figure 6A*.

**Source data 2.** Original files corresponding to *Figure 6A*.

**Source data 3.** Original membranes corresponding to *Figure 6C*.

**Source data 4.** Original files corresponding to *Figure 6C*.

**Figure supplement 1.** Mad2 is required for permanent arrest in a 2-DSB strain.

**Figure supplement 2.** Mad1 and Mad2 are required for permanent arrest in a 2-DSB strain.

**Figure supplement 2—source data 1.** Original membranes corresponding to *Figure 6—figure supplement 2B*.

**Figure supplement 2—source data 2.** Original files corresponding to *Figure 6—figure supplement 2B*.

**Figure supplement 2—source data 3.** Original membranes corresponding to *Figure 6—figure supplement 2C*.

**Figure supplement 2—source data 4.** Original files corresponding to *Figure 6—figure supplement 2C*.

**Figure supplement 3.** Degradation of Ddc2 and Mad2 at 15 hr releases cells from checkpoint arrest.

*Figure 6 continued on next page*

*Figure 6 continued*

**Figure supplement 3—source data 1.** Original membranes corresponding to *Figure 6—figure supplement 3A*.

**Figure supplement 3—source data 2.** Original files corresponding to *Figure 6—figure supplement 3A*.

(*Figure 7A*, *Figure 7—figure supplement 1A and B*), akin to what we observe following Mad1-AID or Mad2-AID depletion (*Figure 6A and B*). In contrast to Bub2-AID, we see that Bfa1-AID degradation did not trigger a significant release from G$_2$/M arrest and did not alter the phosphorylation of Rad53 (*Figure 7B*, *Figure 7—figure supplement 1C–E*). Analysis of cell cycle distribution with DAPI staining showed that neither the inactivation of Bub2 nor Bfa1 led to the accumulation of a significant number of cells with two separate DAPI-staining nuclei, which is indicative of mitotic exit defects (*Figure 7B and D*, *Figure 7—figure supplement 1B, D, and F*). These results imply that although Bub2 and Bfa1 have interdependent functions for MEN signaling, they carry out independent roles in response to DNA damage.

## The location of the second DSB site relative to the centromere affects prolonged arrest

In contrast to the permanent cell cycle arrest we observe in response to 2 DSBs, a recent study using two HO-mediated persistent DSBs showed that cells in fact adapt (*Sadeghi et al., 2022*). One difference between these two 2-DSB systems is the relative position of the DSBs, which might affect how SAC components become engaged, and thus might determine the extent of mitotic arrest. Supporting this, we previously showed that deleting *CEN3* in a strain with a DSB at *MAT* on chromosome III eliminated the Mad2-dependent delay in adaptation, but deleting *CEN3* when the DSB was on chromosome VI had no effect (*Dotiwala et al., 2010*). In our adaptation-defective 2-DSB strain, the DSBs are located at *MAT* (86 kb from *CEN3*) and near *FAB1* (42 kb from *CEN6*). Sadeghi et al. employed two strains, both of which escape prolonged G$_2$/M arrest, with at least one DSB site far from its centromere; at *URA3* (36 kb from its centromere) and *ADH1* (170 kb) or at *MIC2* (32 kb) and *DLD2* (316 kb).

To investigate whether the distance between the second DSB site and the centromere would affect whether cells will remain permanently arrested, we created several strains that contain a second cut site at various distances from the centromere, in addition to the cut site at *MAT* on chromosome III (*Figure 8*). Strain YSL53, which has a second DSB at chromosome V, 86 kb away from *CEN5* (*Lee et al., 1998*), and strain DW417, which has a second DSB 52 kb away from *CEN6* (*Lee et al., 2014*), mostly remained in G$_2$/M arrest 24 hr after the induction of DNA breaks. However, in GEM188, which had a second cut site 230 kb away from the *CEN2*, only 37% of cells remained in G$_2$/M arrest by 24 hr. Thus, the increased distance of the second DSB site to the centromere in GEM188 appears to have led to a less robust triggering of the SAC compared with YSL53 and DW417.

Our previous data have suggested that the involvement of the SAC in prolonging DSB-induced arrest was dependent on centromere sequences on the broken chromosome and involved post-translational modification of chromatin by the Mec1- and Tel1-dependent phosphorylation of the histone H2A (*Dotiwala et al., 2010*). In budding yeast, histone H2B is also targeted by DDC kinases upon DNA damage (*Lee et al., 2014*). To test whether the presence of these chromatin modifications around centromeres would be sufficient to elicit a SAC response, we examined cell cycle progression in strains in which both histone H2A and/or histone H2B genes were mutated to their putative phosphomimetic forms (H2A-S129E and H2B-T129E). We note that although histone H2A-S129E is recognized by an antibody specific for the phosphorylation of histone H2A-S129 (*Eapen et al., 2012*), the mutation to S129E may not be fully phosphomimetic. As shown in *Figure 9*, there was no effect on the growth rate of either the single or the double mutants, suggesting that cells did not experience a SAC-dependent delay in entering mitosis because of these modifications.

## Discussion

Here, we studied how cells maintain cell cycle arrest following DNA damage by using a yeast strain that permanently halts cell cycle with two persistent DNA breaks. We find that most of the DDC signaling proteins must remain active to maintain G$_2$/M arrest, highlighting the importance of

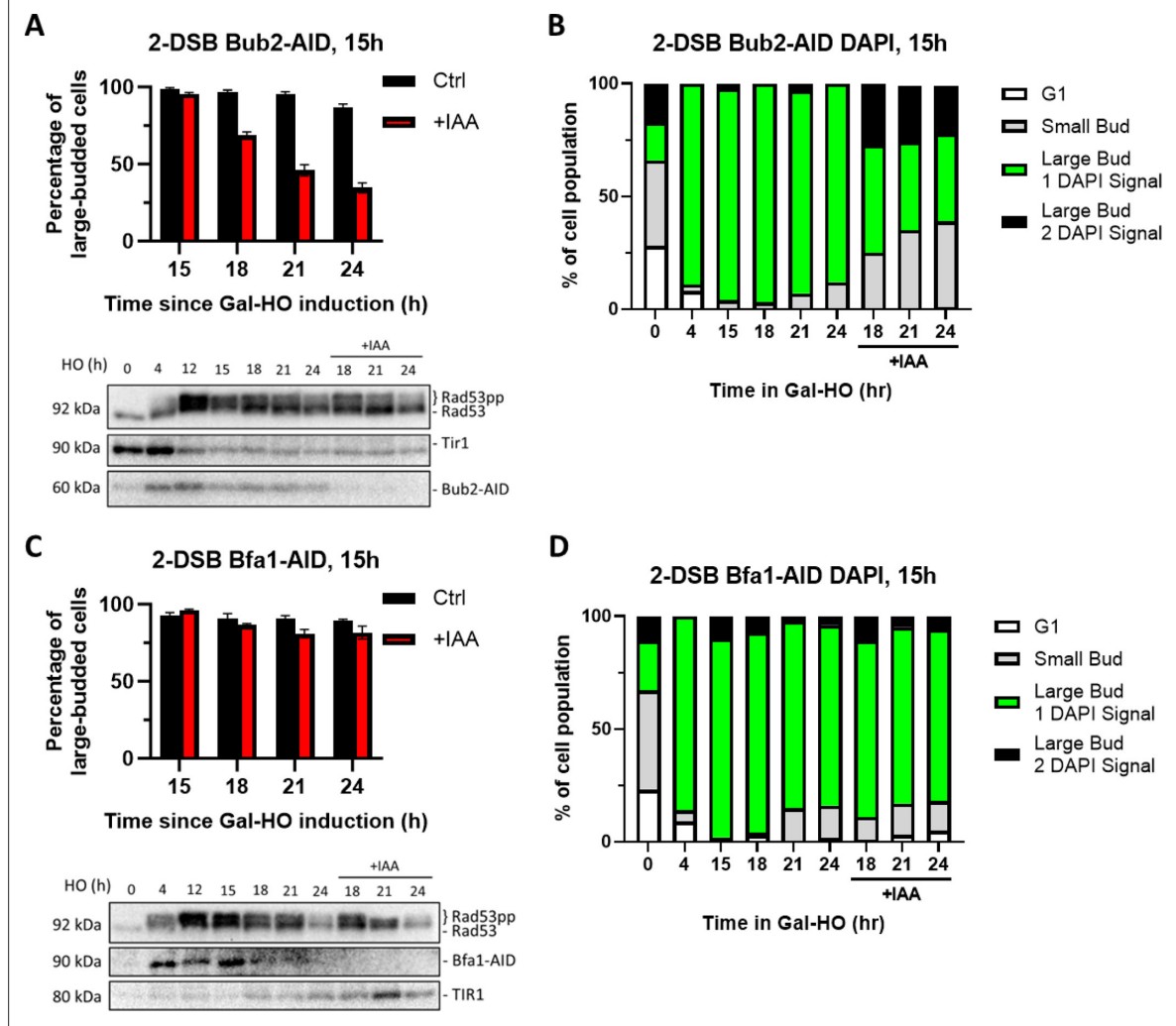

**Figure 7.** Degradation of Bub2 but not Bfa1 at 15 hr releases cells from checkpoint arrest. (**A**) Percentage of G$_2$/M-arrested cells for 2-DSB *BUB2-AID* after HO induction. Data is shown from three trials with standard error of the mean (SEM). Western blot probed with α-Rad53 and α-Myc. α-Rad53 shows both an unphosphorylated protein and multiple phosphorylated species. α-Myc shows Bub2-AID degradation and TIR1-Myc as a loading control. (**B**) Profile of DAPI-stained cells in a 2-DSB *BUB2-AID* strain after HO induction. Liquid cultures were split 15 hr after HO induction and treated with either IAA or ethanol. Cells were scored based on cell morphology and number of DAPI signals. (**C**) Same as (**A**) for 2-DSB *BFA1-AID*. (**D**) Same as (**B**) for 2-DSB *BFA1-AID*.

The online version of this article includes the following source data and figure supplement(s) for figure 7:

**Source data 1.** Original membranes corresponding to *Figure 7A*.

**Source data 2.** Original files corresponding to *Figure 7A*.

**Source data 3.** Original membranes corresponding to *Figure 7C*.

**Source data 4.** Original files corresponding to *Figure 7C*.

**Figure supplement 1.** Bub2 but not Bfa1 is required for prolonged arrest.

**Figure supplement 1—source data 1.** Original membranes corresponding to *Figure 7—figure supplement 1A*.

**Figure supplement 1—source data 2.** Original files corresponding to *Figure 7—figure supplement 1A*.

**Figure supplement 1—source data 3.** Original membranes corresponding to *Figure 7—figure supplement 1C*.

**Figure supplement 1—source data 4.** Original files corresponding to *Figure 7—figure supplement 1C*.

continuous checkpoint signaling in preventing premature mitotic entry and therefore, genome instability (*Figure 10*).

Phosphorylation of Rad53 by Mec1 is tightly linked to cell cycle arrest following DNA damage (*Pellicioli et al., 2001*). While Rad53 is also shown to be targeted by Tel1, deleting *TEL1* did not

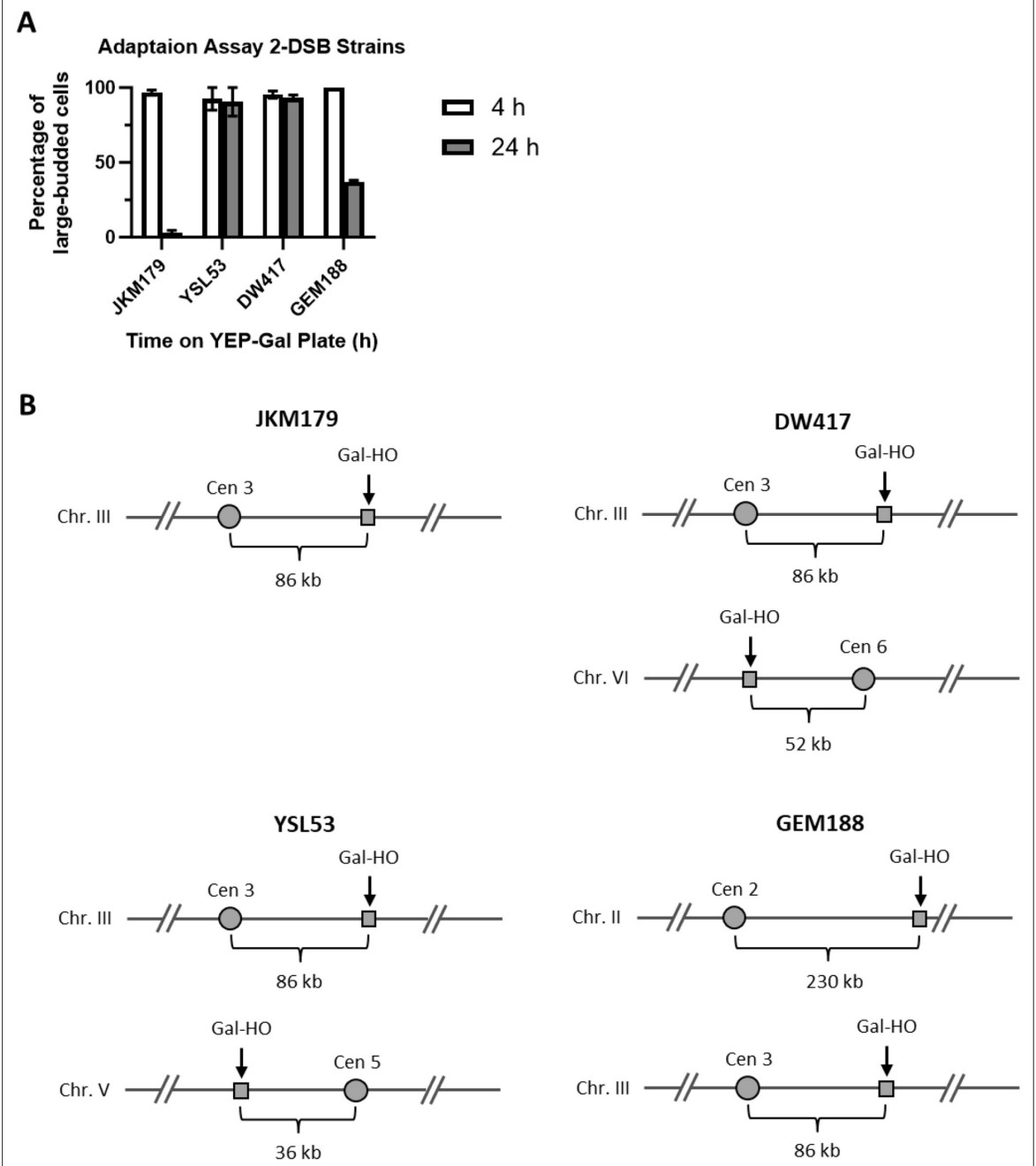

**Figure 8.** Adaptation assay of different 2-DSB strains. (**A**) Adaptation assay of 1-DSB and 2-DSB strains tracking the morphology of 50 G$_1$ cells on a YEP-Gal plate. The percentage of G$_2$/M-arrested cells was shown 4 hr and 24 hr after placement of YEP-Gal plates. JKM179 is a 1-DSB strain with an HO-cut site in the *MAT* locus on chromosome III 86 kb away from the centromere. DW417 is a 2-DSB strain derived from JKM179 with an additional HO-cut site on chromosome VI 52 kb away from the centromere. YSL53 is a 2-DSB strain derived from JKM179 with an additional HO-cut site at the *URA3* locus on chromosome V 36 kb away from the centromere (***Lee et al., 1998***). GEM188 is a 2-DSB strain derived from JKM179 with an additional HO-cut site at *LYS2* on chromosome II 230 kb away from the centromere. (**B**) Cartoon representations of strains showing the location of the HO-cut sites relative to their respective centromeres.

affect the prolonged checkpoint arrest in a 2-DSB strain. This finding supports the previous reports showing that cell cycle arrest in response to enzymatic DNA breaks is largely orchestrated by Mec1, with a minor contribution from Tel1 (***Pellicioli et al., 2001***; ***Vaze et al., 2002***). Here, we add that Mec1 inactivation via Ddc2 depletion, 4 hr after DSB induction, results in rapid resumption of the cell cycle, most likely through Ptc2, Ptc3, Pph3, and Glc7-dependent dephosphorylation and inactivation of the

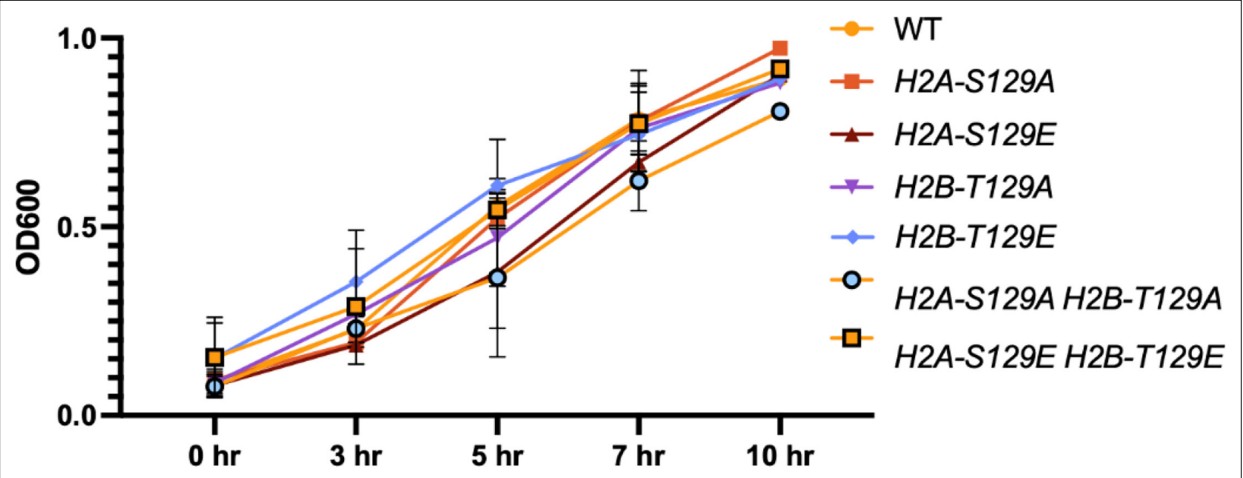

**Figure 9.** Phosphomimetic and non-phosphorylatable mutants of histone H2A and H2B do not affect the growth rate of cells. Growth rate of strains were measured in YPD (2% dextrose) in H2A and H2B mutants for up to 10 hr. Cultures were grown in YPD until they reached an $OD_{600}$ of 0.1. The OD of each strain was then measured at 3, 5, 7, and 10 hr.

DDC signaling protein Rad53 (*Bazzi et al., 2010*; *Leroy et al., 2003*; *O'Neill et al., 2007*). However, the regulation and consequences of Rad53 phosphorylation are apparently more complex, given that cells can re-enter mitosis in the presence of persistent Rad53 phosphorylation when SAC proteins Mad1 and Mad2 are depleted (see below).

Mec1 activation in the S and $G_2$ cell cycle phases is achieved by at least two converging mechanisms; first, through Ddc2 binding to RPA-coated ssDNA created the 5' to 3' resection of the DSB ends, and second, through binding of the 9-1-1 clamp subunit Ddc1 (*Dubrana et al., 2007*; *Navadgi-Patil and Burgers, 2009*; *Zou and Elledge, 2003*; *Melo et al., 2001*). Before Ddc1 can activate Mec1, the 9-1-1 checkpoint clamp must be loaded by the clamp loader, which consists of Rad24-Rfc2-5 (*Melo et al., 2001*). DDC activation largely depends on a functional clamp loader as cells lacking *RAD24* proceed directly into mitosis in response to a single DSB, with only a brief delay (*Aylon and Kupiec, 2003*). If the clamp loader acts just once to load the 9-1-1 clamp and the clamp then slides away from the DSB as DNA is resected, then removal of Rad24 after the checkpoint had been robustly activated should not perturb arrest. However, we find that Rad24 depletion leads to rapid cell cycle resumption in response to 2 DSBs, suggesting that multiple 9-1-1 clamp loading events are required to sustain extended cell cycle arrest. We posit that 9-1-1 clamp may require continuous reloading as the 5' end is being continuously resected by Exo1 and Sgs1-Rmi1-Top3-Dna2 exonucleases (*Zhu et al., 2008*).

The adaptor kinase Rad9, downstream of Mec1 and Tel1, is responsible for scaffolding and activating the effector DDC kinases Rad53 and Chk1 (*Emili, 1998*; *Schwartz et al., 2002*; *Sweeney et al., 2005*). Here, we show that conditional depletion of Rad9 shortly after the induction of 2 DSBs prompts mitotic re-entry and terminates the DDC, evident from rapid Rad53 dephosphorylation. Thus, Rad9 is required for continued maintenance of Rad53 phosphorylation.

Depletion of Rad53-AID 4 hr after inducing 2 DSBs triggers release of cells from $G_2$/M arrest, albeit resumption of mitosis is delayed by 4 hr compared to resumption of mitosis seen when upstream DDC factors Ddc2, Rad9, or Rad24 are depleted. We discovered that this residual delay in mitotic re-entry is dependent on Chk1 signaling. We postulate that Chk1 may delay cell cycle re-entry in the absence of Rad53 by phosphorylating and stabilizing Pds1 (*Agarwal et al., 2003*).

Agreeing with previously published work (*Yam et al., 2020*), we find that deleting *DUN1* resulted in a less robust DDC activation and a shortened $G_2$/M arrest. However, unlike the depletion of other DDC proteins examined in this study, depletion of Dun1-AID 4 hr after DSB induction was not sufficient to promote cell cycle re-entry. It is possible that the transcripts upregulated by Dun1 after DNA damage (*Zhao and Rothstein, 2002*; *Zhou and Elledge, 1993*) are stable for several hours and are sufficient to sustain the arrest even in the absence of Dun1. We also find that cell cycle arrest upon depletion of Dun1 is Chk1-dependent. These results are in consistent with Dun1's role in stabilizing Pds1 through a Chk1-independent mechanism (*Yam et al., 2020*).

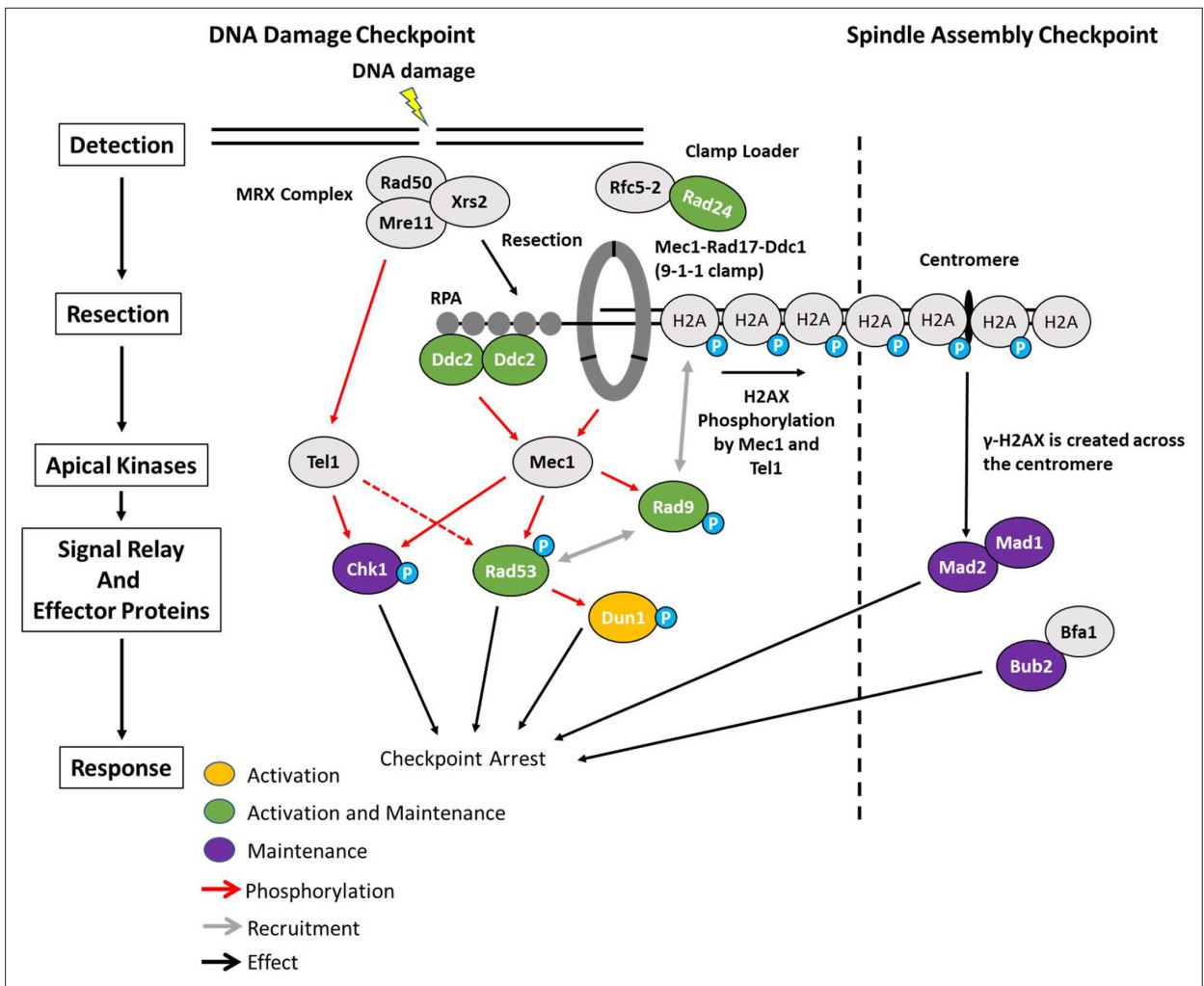

**Figure 10.** Activation and maintenance of checkpoint arrest in response to a DSB. The Mre11-Rad50-Xrs2 (MRX) complex is one of the first complexes recruited to DSBs and initiates the resection of dsDNA to ssDNA. ssDNA is then coated with RPA which recruits Ddc2. Mec1 is the primary kinase responsible for checkpoint arrest in budding yeast and is activated by Ddc2 and Ddc1 from the 9-1-1 clamp. Proteins in green (Ddc2, Rad9, Rad24, and Rad53) were required for the activation and maintenance of checkpoint arrest. While Chk1 was not required for establishment of G$_2$/M arrest, it contributed to the maintenance of arrest. In contrast, Dun1 was required for checkpoint activation but was dispensable 4 hr after DSB induction. Prolonged arrest >24 hr in a 2-DSB strain was dependent on the SAC proteins Mad2, Mad1, and Bub2 as well as the distance between the second HO-cut site and the centromere.

Here, we show that prolonged cell cycle arrest following induction of 2 DSBs becomes independent of DDC proteins Ddc2, Rad9, and Rad53, but dependent on SAC proteins Mad1 and Mad2. Depletion of Mad1 or Mad2 4 hr after DSB induction did not immediately result in cell cycle re-entry, but around 15 hr cells began to resume cell cycle. The timing of cell cycle re-entry for Mad1/2-depleted cells following the induction of 2 DSB is about the same as the timing of cell cycle re-entry in response to a single DNA break in wild-type cells. Even when DDC signaling is artificially upregulated via Ddc2 overexpression, cells re-enter mitosis if Mad2 is depleted. Surprisingly, in the absence of Mad1 or Mad2 cells escaped arrest despite persistent Rad53 hyperphosphorylation, suggesting that cells become 'deaf' to the DDC signal once SAC takes over.

Previous work indicates that SAC proteins contribute to the DNA damage response (**Dotiwala et al., 2010**; **Garber and Rine, 2002**; **Kim and Burke, 2008**). Work from our lab has suggested that γ-H2AX spreading from a DSB to the centromere of the same chromosome might impair kinetochore attachment and thus trigger a SAC response (**Dotiwala et al., 2010**). SAC signaling is not sufficient to elicit a permanent cell cycle arrest in response to a single DSB; however, we find that inducing 2 DSBs, each within 100 kb of its centromere, elicits a SAC-dependent permanent arrest. As strength of

the SAC has a direct relation to the number of unattached kinetochores (*Dick and Gerlich, 2013*), the addition of a second DSB on another chromosome might trigger a stronger SAC response and result in a permanent cell cycle arrest.

The fact that not all combinations of 2 DSBs produce permanent arrest (*Sadeghi et al., 2022*) can be explained by the idea that the distance between the DSB site and its corresponding centromere is an important determinant of the extent of cell cycle arrest. We previously observed that a strain with a single DSB 200 kb away from the centromere had shorter cell cycle arrest compared to a strain with a DSB 86 kb away from the centromere (*Dotiwala et al., 2010*). Here, we provide further evidence suggesting that the distances between DSBs and their corresponding centromeres determine whether SAC will be fully activated to prolong the arrest. Our previous results showed that when *MAD2* was deleted, the length of cell cycle arrest was the same in strains with a single DSB, irrespective of the DSB's distance to its centromere (*Dotiwala et al., 2010*). We suggest that the strains used by *Sadeghi et al., 2022* do not remain permanently arrested because one of the two DSBs in their strains is sufficiently far from its centromere to fully trigger SAC.

In addition to blocking the metaphase to anaphase transition, components of the SAC also block mitotic exit (reviewed by *Matellán and Monje-Casas, 2020*). We examined the role of Bub2/Bfa1 heterodimer, the most upstream components of the MEN pathway (*Matellán and Monje-Casas, 2020*). Much like Mad1 and Mad2, we found that neither Bub2 nor Bfa1 was required for the establishment of cell cycle arrest in response to DNA damage; but our study revealed a surprising result: Bub2, but not its partner Bfa1, is essential to prolong cell cycle arrest, indicating that Bub2 has a Bfa1-independent role.

By using conditional depletion of various proteins that contribute to cell cycle arrest, we show that the establishment, maintenance, and inactivation stages of DNA damage-provoked cell cycle arrest involve different sets of factors following DNA damage. After the DNA damage checkpoint is established, its maintenance proves to be divided into two distinct phases. Arrest up to about 15 hr requires the constant presence of most of the identified DDC proteins, including Ddc2, Rad9, Rad24, and Rad53, with Dun1 playing an important but nonessential role. Although Chk1 was not required either to establish or initially to maintain cell cycle arrest, its absence shortened arrest, most notably when Rad53 was depleted. Surprisingly, neither Ddc2, Rad9, nor Rad53 (and we suggest likely Rad24) are necessary for the prolongation of cell cycle arrest lasting longer than 15–24 hr. Instead, this prolonged arrest is enforced by SAC proteins Mad1, Mad2, and Bub2.

# Materials and methods
## Yeast strain and plasmid construction
All AID-tagged mutant strains were derived from a modified version of strains JKM179. To create the strain with two HO cleavage sites (DW417), an HO-cut site, designated HOcse6, with an adjacent HPH marker was integrated into chromosome VI, 52 kb from the centromere. To create AID strains, we first integrated osTIR1 at *URA3* after digesting plasmid pNHK53 (*Nishimura et al., 2009*) with *Stu*I. To integrate osTIR1-F74G at *URA3*, the plasmid pMK420 (*Yesbolatova et al., 2020*) was digested with *Stu*I. For degron-tagging of DDC proteins, AID-9xMyc (AID) PCR products were generated with mixed oligos with homology to the C-terminal end of the corresponding open-reading frames by using plasmids pKan–AID–9xMyc (pJH2892) or pNat–AID–9xMyc (pJH2899) as templates (*Morawska and Ulrich, 2013*). Deletion of ORFs and insertion of AID tags were introduced with the one-step PCR homology cassette amplification and the standard yeast transformation method (*Wach et al., 1994*). Cas9 editing was done by inserting a gRNA into plasmid bRA90 (*Anand et al., 2017*) and co-transformed into our strain of interest with a donor sequence. Transformants were verified by PCR, western blotting, and sequencing. To create strain GEM188 with 2-DSBs, we inserted a second HO-cut site into JKM179 at *LYS2* locus by CRISPR/Cas9 (*Anand et al., 2017*) using a synthetic DNA template with 117 bp consensus HO recognition site. Non-phosphorylatable and phosphomimetic mutants of H2A were generated in a JKM179 background using CRISPR/Cas9 to target HTA1 and HTA2 genes at serine 129 and 80 nt templates to mutate serine to either alanine (non-phosphorylatable) or glutamic acid (phosphomimetic). H2B mutants were generated in a JKM179 background using CRISPR/Cas9 to target *HTB1* and *HTB2* genes at threonine 129 with 80 nt repair templates to mutate threonine to either alanine or glutamic acid.

The *CEN/ARS* plasmid pFZ052-*pRAD9-AID-\*9Myc::Trp1* (pRAD9-AID) was obtained by digesting the plasmid pFL36.1 (*Lazzaro et al., 2008*) with *SmaI* and *AscI* to excise the *3* HA tag on the C-terminal end of Rad9. A *9xMyc-AID* PCR product generated from the plasmid pJH2892 (pKan-9xMyc-AID) was cut with *AscI* and sticky/blunt end cloned into the *SmaI-AscI*-digested pFL36.1 to add the *9xMyc-AID* tag to the C-terminal end of Rad9. pRad9-AID was retained by growing cells in a Trp- media with 2% raffinose. The primers used for strain and plasmid creation are listed in the Key Resources Table, *Supplementary file 1a*, and *Supplementary file 1b*. Plasmids are listed in the Key Resources Table.

## Culturing conditions, HO expression, and auxin treatment

Strains containing degron fusions and galactose-inducible HO were cultured using standard procedures. Briefly, a single colony grown on a YEPD plate (1% yeast extract, 2% peptone, 2% dextrose, 2.5% agar) was inoculated in 5 ml YEP-lactate (YEP containing 3% lactic acid) and was grown for ~15 hr at 30°C with agitation. Next day, the overnight culture was used to inoculate a 500–100 ml of YEP-lactate culture such that the cell density reached an $OD_{600}$ of 0.5 the following day. After harvesting 15 ml liquid culture before treatment, HO expression was induced by galactose treatment with a 2% final concentration. Then, cultures were split either at 4 hr or 15 hr following induction with galactose. The split cultures were treated either with IAA or 5-Ph-IAA or an equivalent volume of 200 proof ethanol. IAA (Sigma-Aldrich, I3750) was resuspended in ethanol for a 500 mM stock and used at a 1 mM final concentration both for liquid media and for agar plates. 5-Ph-IAA was dissolved in ethanol for a 1 mM stock and used at a final concentration of 1 µM. 15 ml of liquid culture were harvested at various timepoints and prepared for microscopy or western blot analysis as described below.

To measure growth rate, strains were grown in 5 ml of YEPD with 2% dextrose at 30°C with an initial $OD_{600}$ of 0.1. The OD was measured 3, 5, 7, and 10 hr after the initial OD measurement using a Thermo Scientific NanoDrop 2000c Spectrophotometer. To measure the OD, 50 µl of culture was added to 950 µl of fresh YPD in a cuvette at a dilution of 1:20.

## TCA protein extraction

Protein extracts were prepared for western blot analysis by TCA extraction protocol as previously explained (*Miller-Fleming et al., 2014*). Briefly, 15 ml of liquid culture was spun down and the media was discarded. Harvested cells were incubated on ice in 1.5 ml microcentrifuge tubes with 20% TCA for 20 min. Cells were washed with acetone and the pellet was air-dried. 200 µl of MURBs buffer (50 mM sodium phosphate, 25 mM MES, 3 M urea, 0.5% 2-mercaptoethanol, 1 mM sodium azide, and 1% SDS) was added to each sample with acid-washed glass beads. Cells were lysed by mechanical shearing with glass beads for 2 min. Crude cell lysates were harvested by poking a hole in the bottom of the 1.5 ml microcentrifuge tube and spinning the tubes on a 15 ml conical tube. Samples were boiled at 95°C for 10 min prior to loading on SDS-PAGE.

## Western blotting

8–20 µl of denatured protein samples prepared by TCA extraction were loaded onto a 10% or 8% SDS-PAGE gels. Proteins were separated by applying constant voltage at 90 V until the 37 kDa protein standard band reached the bottom of the gel. Gels were transferred to an Immun-Blot PVDF membrane (Bio-Rad) using a wet transfer apparatus set to 100 V constant voltage for 1 hr at 4°C. Membranes were then blocked with OneBlock blocking buffer (Genesee Scientific, Cat# 20-313) for 1 hr at room temperature, After three 10 min washes with 1× TBS-T, blots were incubated with either anti-Myc [9E11] (Abcam, ab56) to detect TIR1 and AID fusions, anti-Rad53 [EL7. E1] (Abcam, ab166859), anti-Pgk1 (Abcam, ab30359), or anti-Rad9 (*Usui et al., 2009*) for 1 hr at room temperature or at 4°C overnight. Blots were washed three times with 1× TBS-T and incubated with anti-mouse HRP (GE Healthcare, Cat# NXA931) or anti-rabbit HRP secondary antibody (Sigma-Aldrich, Cat# A6154) for 1 hr at room temperature. After washing the membranes three times with 1× TBS-T, Amersham ECL Prime Western Blotting Detection Reagent was added to fully coat the blots and left to incubate for 5 min at room temperature with gentle agitation. Blots were imaged using a Bio-Rad ChemiDoc XR+ imager and prepared for publication using ImageLab 6.1 software (Bio-Rad) and Adobe Photoshop CC 2017. The reagents used are listed in the Key Resources Table.

## Microscopy, DAPI staining, and cell morphology determination

Aliquots from YEP-Lac cultures were taken either 4 hr or 15 hr after adding galactose, diluted 20-fold in sterile water, and plated on a YEP-Agar with 2% galactose with or without 1 mM IAA or 1 µM 5-Ph-IAA. Cells were counted on a light microscope with a ×10 objective, examined, and binned into three categories: unbudded, small buds, and $G_2$/M-arrested cells with large buds. For each timepoint, >250 cells were analyzed. For DAPI staining, 450 µl of culture was added to 50 µl of 37% formaldehyde and incubated in the chemical hood at room temperature for 20 min. Samples were spun down at 8000 rpm for 5 min and washed with 1× PBS three times. Cells were resuspended in 50 µl of DAPI mounting media (VECTASHIELD Antifade Mounting Medium with DAPI H-1200-10) and incubated at room temperature for 10 min, away from direct light. The samples were imaged by using a Nikon Ni-E upright microscope equipped with a Yokogawa CSU-W1 spinning-disk head, an Andor iXon 897U EMCCD camera, Nikon Elements AR software, a ×60 oil immersion objective, and a 358 nm laser. Fifteen z-stacks with a thickness of 0.3 µm were collected per image. In the morphology assays, at least three biological replicates were used for each strain.

## Adaptation and auxin plating assays

We performed adaptation assays as previously described (*Eapen et al., 2012*; *Lee et al., 1998*). Cells grown in YEP-Lac overnight were diluted 20-fold in sterile water and plated on a YEP-agar plate containing 2% galactose. Using micromanipulation, 50 $G_1$ cells were isolated and positioned in a grid followed by incubation at 30°C. To quantify the percentage of adapted cells, the number of cells that re-entered cell cycle and grew to a microcolony (3+ cells) after 24 hr was divided by the total number of cells. For auxin plating assays, damage was induced in a YEP-Lac liquid culture by adding galactose at a final concentration of 2%, as described above. Cells were then transferred onto YEP-agar plates containing 2% galactose and 1 mM IAA or 1 µM 5-Ph-IAA 4 hr or 15 hr after adding galactose. For each timepoint, >250 cells were scored and categorized as described above for the adaptation assay from at least three biological replicates.

## Quantification and data analysis

Graphs were prepared using GraphPad Prism 10 (Dotmatics). Statistical analysis for differences in the percentage of large budded ($G_2$/M-arrested) cells at different timepoints listed in *Table 1* was done using a one-way ANOVA in GraphPad Prism 10. Protein quantification of Ddc2-myc blots was done using ImageLab 6.1 (Bio-Rad). To categorize DAPI-stained cells based on their morphology and number of DAPI signals, images were captured as described above and viewed using ImageJ with the Fiji addon.

# Acknowledgements

We thank Helle Ulrich for the AID and TIR1 plasmid and Masato Kanemaki for the TIR1(F74G) plasmid. We also thank Nikita Alimov, Jessie Ang, and Astré Bouchier for their assistance.

# Additional information

## Funding

| Funder | Grant reference number | Author |
|---|---|---|
| National Institutes of Health | R35 GM127029 | Felix Y Zhou<br>David P Waterman<br>Marissa Ashton<br>Vinay V Eapen<br>James E Haber |
| National Institutes of Health | TM32GM007122 | Felix Y Zhou<br>David P Waterman<br>Suhaily Caban-Penix |
| Howard Hughes Medical Institute | | Vinay V Eapen |

| Funder | Grant reference number | Author |
| --- | --- | --- |
| National Institutes of Health | F32-GM145156 | Gonen Memisoglu |
| National Institute of General Medical Sciences | 5T32GM139798 | Marissa Ashton |

The funders had no role in study design, data collection and interpretation, or the decision to submit the work for publication.

## Author contributions

Felix Y Zhou, Resources, Data curation, Formal analysis, Validation, Investigation, Visualization, Methodology, Writing – original draft, Writing – review and editing; David P Waterman, Conceptualization, Resources, Data curation, Formal analysis, Investigation, Methodology, Writing – original draft, Writing – review and editing; Marissa Ashton, Resources, Data curation, Formal analysis, Investigation; Suhaily Caban-Penix, Resources, Data curation, Investigation; Gonen Memisoglu, Resources, Data curation, Investigation, Writing – review and editing; Vinay V Eapen, Conceptualization, Data curation, Formal analysis, Investigation, Methodology; James E Haber, Conceptualization, Supervision, Writing – original draft, Project administration, Writing – review and editing

## Author ORCIDs

Felix Y Zhou ⓘ https://orcid.org/0000-0002-8619-7160
James E Haber ⓘ https://orcid.org/0000-0002-1878-0610

Reviewer #1 (Public review): https://doi.org/10.7554/eLife.94334.3.sa1
Reviewer #2 (Public review): https://doi.org/10.7554/eLife.94334.3.sa2
Reviewer #3 (Public review): https://doi.org/10.7554/eLife.94334.3.sa3
Author response https://doi.org/10.7554/eLife.94334.3.sa4

---

# Additional files

## Supplementary files

• Supplementary file 1. Strains and Primers used in this study. (a) Strains used in this study. (b) Primers used in this study.

• MDAR checklist

## Data availability

Data is available on Dryad: https://doi.org/10.5061/dryad.sj3tx96dv.

The following dataset was generated:

| Author(s) | Year | Dataset title | Dataset URL | Database and Identifier |
| --- | --- | --- | --- | --- |
| Zhou FY, Waterman DP, Caban-Penix S, Eapen VV, Haber JE | 2024 | Prolonged Cell Cycle Arrest in Response to DNA damage in Yeast Requires the Maintenance of DNA Damage Signaling and the Spindle Assembly Checkpoint | http://dx.doi.org/10.5061/dryad.sj3tx96dv | Dryad Digital Repository, 10.5061/dryad.sj3tx96dv |

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

# Appendix 1

## Appendix 1—key resources table

| Reagent type (species) or resource | Designation | Source or reference | Identifiers | Additional information |
|---|---|---|---|---|
| Antibody | Anti-Rad53; (mouse monoclonal) | Abcam | Cat# ab166859; RRID:AB_2801547 | WB (1:1000) |
| Antibody | Anti-Rad53; (rabbit polyclonal) | Abcam | ab104232 | WB (1:1000) |
| Antibody | Anti-Myc; (mouse monoclonal) | Abcam | Cat# ab16918; RRID:AB_30256 | WB (1:1000) |
| Antibody | Anti-Pgk1; (mouse monoclonal) | Abcam | Cat# ab32; RRID:AB_30359 | WB (1:5000) |
| Antibody | Anti-Rad9; (rabbit polyclonal) | *Usui et al., 2009* | N/A | WB (1:4000) |
| Antibody | ECL TH Anti-mouse IgG horseradish peroxidase from sheep | GE Healthcare | NXA931V Lot 16937010 | WB (1:10,000) |
| Antibody | ECL TH Anti-rabbit IgG horseradish peroxidase from donkey | GE Healthcare | NA934V lot 6969611 | WB (1:10,000) |
| Chemical compound, drug | Indole-3-acetic acid | Sigma-Aldrich | I3750-25G-A | 1 mM |
| Chemical compound, drug | 5-Ph-IAA | Sigma-Aldrich | SML3574-25MG | 1 µM |
| Chemical compound, drug | Formaldehyde | Sigma-Aldrich | 47608 | 4% |
| Chemical compound, drug | VECTASHIELD Antifade Mounting Medium with DAPI | Vector Laboratories | Cat# H-1200 | 1 µg/ml |
| Chemical compound, drug | Prometheus Protein Biology Products 20–313 OneBlock Western-CL Blocking Buffer, For Chemiluminescent Blots | Genesee Scientific | Cat# 20-313 | Blocking buffer for western blots |
| Commercial assay or kit | ECL Prime Western Blotting System | MilliporeSigma | GERPN2232 | |
| Strain, strain background (*Saccharomyces cerevisiae*) | *cerevisiae*: strain background S228c | See *Supplementary file 1a* for full strain list | N/A | Strains used in this study |
| Strain, strain background (*S. cerevisiae*) | JKM179 | *Lee et al., 1998* | Yeast strain | *MATα ade1 leu2-3 lys5 trp1 ::hisG ura3-52 hoΔ hmlΔ::ADE1 hmr Δ::ADE1 ade3:: GAL::HO* |
| Strain, strain background (*S. cerevisiae*) | DW184 | This study | Yeast strain | *TIR1-myc6::URA3* |
| Strain, strain background (*S. cerevisiae*) | DW417 | This study | Yeast strain | *HOcse6::HPH TIR1 -myc6::URA3* |
| Strain, strain background (*S. cerevisiae*) | DW418 | This study | Yeast strain | *Ddc2-AID*–9xMyc::KAN* |
| Strain, strain background (*S. cerevisiae*) | DW419 | This study | Yeast strain | *Rad9-AID*–9xMyc::KAN* |
| Strain, strain background (*S. cerevisiae*) | DW420 | This study | Yeast strain | *Rad24-AID*–9xMyc::KAN* |
| Strain, strain background (*S. cerevisiae*) | DW421 | This study | Yeast strain | *Rad53-AID*–9xMyc::KAN* |
| Strain, strain background (*S. cerevisiae*) | DW426 | This study | Yeast strain | *chk1Δ::NAT* |
| Strain, strain background (*S. cerevisiae*) | DW647 | This study | Yeast strain | *Ddc2-AID*–9xMyc::KAN chk1Δ::NAT* |
| Strain, strain background (*S. cerevisiae*) | DW427 | This study | Yeast strain | *Rad9-AID*–9xMyc::KAN chk1Δ::NAT* |
| Strain, strain background (*S. cerevisiae*) | DW428 | This study | Yeast strain | *Rad24-AID*–9xMyc::KAN chk1Δ::NAT* |

*Appendix 1 Continued on next page*

*Appendix 1 Continued*

| Reagent type (species) or resource | Designation | Source or reference | Identifiers | Additional information |
|---|---|---|---|---|
| Strain, strain background (*S. cerevisiae*) | DW429 | This study | Yeast strain | *Rad53-AID\*–9xMyc::KAN chk1Δ::NAT* |
| Strain, strain background (*S. cerevisiae*) | DW625 | This study | Yeast strain | *dun1Δ::KAN* |
| Strain, strain background (*S. cerevisiae*) | DW626 | This study | Yeast strain | *Dun1-AID\*–9xMyc::KAN* |
| Strain, strain background (*S. cerevisiae*) | DW641 | This study | Yeast strain | *Dun1-AID\*–9xMyc::KAN chk1Δ::NAT* |
| Strain, strain background (*S. cerevisiae*) | FZ009 | This study | Yeast strain | *Mad2\*–9xMyc-AID::NAT* |
| Strain, strain background (*S. cerevisiae*) | FZ010 | This study | Yeast strain | *Ddc2-AID\*–9xMyc ::KAN Mad2\*–9xMyc -AID::NAT* |
| Strain, strain background (*S. cerevisiae*) | DW455 | This study | Yeast strain | *mad2Δ::KAN* |
| Strain, strain background (*S. cerevisiae*) | GM180 | **Memisoglu et al., 2019** | Yeast strain | *pGal::Ddc2::LEU2* |
| Strain, strain background (*S. cerevisiae*) | DW648 | This study | Yeast strain | *HOcse6::HPH pGal:: Ddc2::LEU2* |
| Strain, strain background (*S. cerevisiae*) | DW649 | This study | Yeast strain | *HOcse6::HPH pGal:: Ddc2::LEU2 mad2Δ::NAT* |
| Strain, strain background (*S. cerevisiae*) | DW642 | This study | Yeast strain | *HOcse6::HPH Ddc2-AID\*–9xMyc::KAN* |
| Strain, strain background (*S. cerevisiae*) | DW643 | This study | Yeast strain | *HOcse6::HPH Rad9-AID\*– 9xMyc::KAN* |
| Strain, strain background (*S. cerevisiae*) | DW644 | This study | Yeast strain | *HOcse6::HPH Rad24-AID\*– 9xMyc::KAN* |
| Strain, strain background (*S. cerevisiae*) | DW645 | This study | Yeast strain | *HOcse6::HPH Rad53-AID\*– 9xMyc::KAN* |
| Strain, strain background (*S. cerevisiae*) | DW650 | This study | Yeast strain | *pGal::Ddc2::LEU2 mad2Δ::NAT* |
| Strain, strain background (*S. cerevisiae*) | GM539 | **Memisoglu et al., 2019** | Yeast strain | *Ddc2-9xMyc::KAN* |
| Strain, strain background (*S. cerevisiae*) | FZ001 | This study | Yeast strain | *MATα HOcse6::HPH* |
| Strain, strain background (*S. cerevisiae*) | JY542 | This study | Yeast strain | *HOcse6::HPH tel1Δ::KAN* |
| Strain, strain background (*S. cerevisiae*) | FZ024 | This study | Yeast strain | *HOcse6::HPH Rad9-AID\*– 9xMyc::KAN* |
| Strain, strain background (*S. cerevisiae*) | FZ025 | This study | Yeast strain | *HOcse6::HPH Rad24-AID\*– 9xMyc::KAN* |
| Strain, strain background (*S. cerevisiae*) | FZ026 | This study | Yeast strain | *HOcse6::HPH Rad53-AID\*– 9xMyc::KAN* |
| Strain, strain background (*S. cerevisiae*) | FZ173 | This study | Yeast strain | *HOcse6::HPH Rad24-AID\*– 9xMyc::KAN TIR1(F74G)::URA3* |
| Strain, strain background (*S. cerevisiae*) | FZ174 | This study | Yeast strain | *HOcse6::HPH Rad9-AID\*– 9xMyc::KAN TIR1(F74G)::URA3* |
| Strain, strain background (*S. cerevisiae*) | FZ175 | This study | Yeast strain | *HOcse6::HPH Rad53-AID\*–9xMyc::KAN TIR1(F74G):: URA3* |
| Strain, strain background (*S. cerevisiae*) | YSL53 | **Lee et al., 1998** | Yeast strain | *HOcse5::URA3* |
| Strain, strain background (*S. cerevisiae*) | GEM188 | This study | Yeast strain | *HOcse2::LYS2* |

*Appendix 1 Continued on next page*

*Appendix 1 Continued*

| Reagent type (species) or resource | Designation | Source or reference | Identifiers | Additional information |
|---|---|---|---|---|
| Strain, strain background (*S. cerevisiae*) | FZ201 | This study | Yeast strain | *Rad9-AID\*–9xMyc::KAN pRAD9-AID\*–9xMyc* |
| Strain, strain background (*S. cerevisiae*) | FZ155 | This study | Yeast strain | *bfa1Δ::KAN* |
| Strain, strain background (*S. cerevisiae*) | yMA11 | This study | Yeast strain | H2A-S129A H2B-T129A |
| Strain, strain background (*S. cerevisiae*) | yMA12 | This study | Yeast strain | H2AS129E H2B-T129E |
| Strain, strain background (*S. cerevisiae*) | yMA13 | This study | Yeast strain | H2B-T129A |
| Strain, strain background (*S. cerevisiae*) | yMA14 | This study | Yeast strain | H2B-T129E |
| Strain, strain background (*S. cerevisiae*) | yBL257 | This study | Yeast strain | H2A-S129E |
| Strain, strain background (*S. cerevisiae*) | yBL259 | This study | Yeast strain | H2A-S129A |
| Strain, strain background (*S. cerevisiae*) | FZ062 | This study | Yeast strain | *Mad1\*–9xMyc-AID::NAT* |
| Strain, strain background (*S. cerevisiae*) | FZ165 | This study | Yeast strain | *Bfa1\*–9xMyc-AID::NAT* |
| Strain, strain background (*S. cerevisiae*) | FZ167 | This study | Yeast strain | *Bub2\*–9xMyc-AID::NAT* |
| Sequence-based reagent | GAT1p1B | This paper | PCR primers | GCTCAGTGTGCGTTATGCTT |
| Sequence-based reagent | GAT1p2B | This paper | PCR primers | TTCAGGTCTCGGTTGCTCTT |
| Sequence-based reagent | VE162 Ddc2-AID For | This paper | PCR primers | ATCTAACCACACTAGAGGAGGCCGATTCATTATATATCTCAATGGGACTGCCTAAAGATCCAGCCAAACCTCC |
| Sequence-based reagent | VE163 Ddc2-AID Rev | This paper | PCR primers | ATTACAAGGTTTCTATAAAGCGTTGACATTTTCCCCTTTTGATTGTTGCCCAGTATAGCGACCAGCATTCACATAC |
| Sequence-based reagent | DW217 Rad9-AID 1 F | This paper | PCR primers | GGTTTTCACGATGATATTACGGACAATGATATATACAACACTATTTCTGAGGTTAGACCTAAAGATCCAGCCAAACCTCC |
| Sequence-based reagent | DW218 Rad9-AID 1 R | This paper | PCR primers | CTAAATTTTTTTTTATTTAATCGTCCCTTTCTATCAATTATGAGTTTATATATTTTTATAATTCAGTATAGCGACCAGCATTCACATAC |
| Sequence-based reagent | DW208 Rad24-AID 1 F | This paper | PCR primers | CAGATTCAGATCTGGAAATACTCCCTAAAGATCCAGCCAAACCTCC |
| Sequence-based reagent | DW209 Rad24-AID 1 R | This paper | PCR primers | GTGGAATATTTCCTGGGGTTTTCTCGTCAAATTTAAAGAGTAAAAAGCCTAAAGATCCAGCCAAACCTCC |
| Sequence-based reagent | DW199 Rad53AID 1 F | This paper | PCR primers | GGTTAAAAGGGCAAAATTGGACCAAACCTCAAAAGGCCCCGAGAATTTGCAATTTTCGCCTAAAGATCCAGCCAAACCTCC |
| Sequence-based reagent | DW200 Rad53AID 1 R | This paper | PCR primers | CCATCTTCTCTCTTAAAAAGGGGCAGCATTTTCTATGGGTATTTGTCCTTGGCAGTATAGCGACCAGCATTCACATAC |
| Recombinant DNA reagent | pKan-9xMyc-AID | *Morawska and Ulrich, 2013* | pJH2892 | Backbone: pSM409 |

*Appendix 1 Continued on next page*

*Appendix 1 Continued*

| Reagent type (species) or resource | Designation | Source or reference | Identifiers | Additional information |
|---|---|---|---|---|
| Recombinant DNA reagent | pNAT-9xMyc-AID | *Morawska and Ulrich, 2013* | pJH2899 | Backbone: pSM409 |
| Recombinant DNA reagent | sTIR1::*URA3* | *Nishimura et al., 2009* | pNHK53 | |
| Recombinant DNA reagent | GAL-DDC2 | *Paciotti et al., 2000* | pML100 | Backbone: pML95 |
| Recombinant DNA reagent | ADH1-OsTIR1(F74G) | *Yesbolatova et al., 2020* | pMK420 | |
| Recombinant DNA reagent | bRA90 | *Anand et al., 2017* | bRA90 | |
| Recombinant DNA reagent | bG059 | This study | bG059 | Backbone: bRA90 |
| Recombinant DNA reagent | bG060 | This study | bG060 | Backbone: bRA90 |
| Recombinant DNA reagent | pRad9-3HA | *Lazzaro et al., 2008* | pFL36.1 | Backbone: pRS306 |
| Recombinant DNA reagent | pRad9-9xMyc-AID | This study | pFZ052 | Backbone: pRS306 |
| Recombinant DNA reagent | pBL15 – HTA1 gRNA1 | This study | pBL15 | Backbone: BRA89 |
| Recombinant DNA reagent | pBL16 – HTA2 gRNA2 | This study | pBL16 | Backbone: BRA89 |
| Recombinant DNA reagent | pKL004 – HTB1 gRNA1 | This study | pKL004 | Backbone: BRA89 |
| Recombinant DNA reagent | pKL005 – HTB2 gRNA1 | This study | pKL005 | Backbone: BRA89 |
| Software, algorithm | Prism 7.00 | GraphPad Software, Inc. | N/A | |
| Software, algorithm | Image Lab | Bio-Rad | N/A | |
| Software, algorithm | FiJi | ImageJ | N/A | |
| Gene (*S. cerevisiae*) | Ddc2 | Saccharomyces Genome Database | Systematic name YDR499W | |
| Gene (*S. cerevisiae*) | Rad9 | Saccharomyces Genome Database | Systematic name YDR217C | |
| Gene (*S. cerevisiae*) | Rad24 | Saccharomyces Genome Database | Systematic name YER173W | |
| Gene (*S. cerevisiae*) | Rad53 | Saccharomyces Genome Database | Systematic name YPL153C | |
| Gene (*S. cerevisiae*) | Chk1 | Saccharomyces Genome Database | Systematic name YBR274W | |
| Gene (*S. cerevisiae*) | Dun1 | Saccharomyces Genome Database | Systematic name YDL101C | |
| Gene (*S. cerevisiae*) | Tel1 | Saccharomyces Genome Database | Systematic name YBL088C | |
| Gene (*S. cerevisiae*) | Mad2 | Saccharomyces Genome Database | Systematic name YJL030W | |
| Gene (*S. cerevisiae*) | Mad1 | Saccharomyces Genome Database | Systematic name YGL086W | |
| Gene (*S. cerevisiae*) | Bub2 | Saccharomyces Genome Database | Systematic name YMR055C | |
| Gene (*S. cerevisiae*) | Bfa1 | Saccharomyces Genome Database | Systematic name YJR053W | |

