## [Editor Report · eLife Assessment]

This is an **important** study on the damage-induced checkpoint maintenance and termination in budding yeast that provides novel and **convincing** evidence for a role of the spindle assembly checkpoint and mitotic exit network in halting the cell cycle after prolonged arrest in response to irreparable DNA double-strand breaks (DSBs). The study identifies particular components from these checkpoints that are specifically required for the establishment and/or the maintenance of a cell cycle block triggered by such DSBs. The authors propose an interesting model for how these different checkpoints intersect and crosstalk for timely resumption of cell cycling even without repairing DNA damage that has been revised by addressing the bulk of the reviewers' comments to the first version of the manuscript.

---

## [Referee Report · Reviewer #1 (Public review)]

Summary:

In their manuscript, Zhou et al. analyze the factors controlling the activation and maintenance of a sustained cell cycle block in response to persistent DNA DSBs. By conditionally depleting components of the DDC using auxin-inducible degrons, the authors verified that some of them are only required for the activation (e.g., Dun1) or the maintenance (e.g., Chk1) of the DSB-dependent cell cycle arrest, while others such as Ddc2, Rad24, Rad9 or Rad53 are required for both processes. Notably, they further show that after a prolonged arrest (>24 h) in a strain carrying two DSBs, the DDC becomes dispensable and the mitotic block is then maintained by SAC proteins such as Mad1, Mad2 or the mitotic exit network (MEN) component Bub2.

Strengths:

The manuscript dissects the specific role of different components of the DDC and the SAC during the induction of a cell cycle arrest induced by DNA damage, as well as their contribution for the short-term and long-term maintenance of a DNA DSB-induced mitotic block. Overall, the experiments are well described and properly executed, and the data in the manuscript are clearly presented. The conclusions drawn are generally well supported by the experimental data. Their observations contribute to drawing a clearer picture of the relative contribution of these factors to the maintenance of genome stability in cells exposed to permanent DNA damage.

Weaknesses:

The main weakness of the study is that it is fundamentally based on the use of the auxin-inducible degron (AID) strategy to deplete proteins. This widely used method allows an efficient depletion of proteins in the cell. However, the drawback is that a tag is added to the protein, which can affect the functionality of the targeted protein or modify its capacity to interact with others. In fact, three of the proteins that are depleted using the AID systems are shown to be clearly hypomorphic, and hence their capacity to induce a strong checkpoint response might be compromised. A corroboration of at least some of the results using an alternative manner to eliminate the proteins would help to strengthen the conclusions of the manuscript.

---

## [Referee Report · Reviewer #2 (Public review)]

Summary:

The manuscript analyzes and attempts to discriminate genetic requirements for DNA damage-induced cell cycle checkpoint induction, maintenance, and adaptation in budding yeast bearing one or two unrepairable DNA double strand breaks using auxin-induced degradation (AID) of key DNA damage response (DDR) factors. The study paid particular attention to solving a puzzle regarding how yeasts bearing two unrepaired DNA breaks fail to engage in "adaptation" whereas those with a single unrepairable break eventually resume cell cycling after a prolonged (up to 12 h) G2 arrest.

The key findings are: 1. Genetic requirements for the entry and the maintenance of DDC are separable. For instance, Dun1 is partially required for the entry but not the DDC maintenance whereas Chk1 is only required for maintenance. 2. Cells with two unrepairable breaks respond to DDR only up to a certain time (~12-15 h post damage) and beyond this point, depend on spindle assembly checkpoint (SAC) and mitotic exit network (MEN) to halt cell cycling. 3. The authors also propose an interesting concept that the location of DNA breaks and their distance to centromeres are important factors dictating the effect of SAC/MEN on the duration of cell cycle arrest after prolonged arrest (and cells become "deaf" to persistent arrest signals) and yeast's adaptability following DNA damage. The results provide most compelling evidence to date on the role of SAC/MEN in DNA damage response and cell cycle arrest albeit its impact might be limited to the handful of model systems due to the vastly different centromeric elements and far larger chromosome sizes in metazoan cells. The study albeit briefly discussed the basis of transitions from entry, maintenance, and adaptation (ex. changes in centromeric architectures), it does not offer detailed explanations or a testable hypothesis to this topic.

Overall, the conclusion of the study is well supported by the elegant set of genetic experimental data and employed multiple readouts on DDC factor depletion on checkpoint integrity and cell cycle status. Although the study simply measures Rad53 phosphorylation as the primary metric to assess checkpoint status, it successfully demonstrated how the signaling is modified through the different stages and that eventually cells become recalcitrant to DDC signaling after a prolonged arrest. The results are clear, and rigorously tested and carefully interpreted with good discussion on the possible limitations. The revision provided detailed responses to the reviewers' comments and addressed a few key concerns, one of which is universally raised by the reviewers on the full functionality of AID tagged DDC factors, by simply expressing excess Rad9-AID to restore more normal looking checkpoint response. It will be interesting if the excess expression of other DDC factors could overcome suboptimal checkpoints in cells after 24 h post damage.

---

## [Referee Report · Reviewer #3 (Public review)]

Summary:

The DNA damage checkpoint (DDC) inhibits the metaphase-anaphase transition to repair various types of DNA damage, including DNA double strand breaks (DSBs). One irreparable DSB can maintain the DDC for 12-15 hours in yeast, after which the cells resume the cell cycle. If there are two DSBs, the DDC is maintained for at least 24 hours. In this study, the authors take advantage of this tighter DDC to investigate whether the best-known proteins involved in establishing the DDC are also responsible for its long-term maintenance during irreparable DSBs. They do this by cleverly degrading such proteins after DSB formation. They show that most, but not all, DDC proteins maintain the cell cycle block. Interestingly, DDC proteins become dispensable after 15 hours and the block is then maintained by spindle assembly checkpoint (SAC) proteins.

Strengths:

The authors have engineered a tight yeast system to study DDC shutdown after irreparable DSBs and used it to address whether checkpoint proteins (DDC and SAC) contribute to the long-term maintenance of DSB-mediated G2/M block. The different roles of Ddc2, Chk1 and Dun1 are interesting, while the fact that SAC overtakes DDC after 15 hours is intriguing and highlights how DSBs near and far from centromeres can have a profound impact on cell adaptation to DSBs. In their revision, the authors have now improved the Rad9-AID methodology to place Rad9 in the context of DDC adaptation, as well as widening the association between adaptation and proximity to centromeres.

Weaknesses:

Some of the results they present essentially confirm their own previous findings, albeit with a tighter strain design for long-term arrest. Conclusions about the maintenance of G2/M in several mutant combinations could have been strengthened by adding simple microscopy experiments with DAPI staining. No clear mechanism for how depletion of Bub2, but not Bfa1, can relieve the G2/M (metaphase) block is given.

---

## [Author Response]

The following is the authors’ response to the original reviews.

**Reviewer #1 (Recommendations For The Authors):**
To hopefully contribute to more strongly support the conclusions drawn by the authors, I am including a series of concerns regarding the manuscript, as well as some suggestions that could be useful to address these issues:(1) The main results of this study derive from the use of auxin-inducible degron (AID)-tagged proteins. Despite the great advantages of the AID strategy to conditionally deplete proteins, the AID tag can affect the normal function of a protein. In fact, some of the AID-labeled DDC components generated in this work are shown to be hypomorphic. Hence, the manuscript would have benefited from the additional confirmation of some of the observations using a different way to eliminate the proteins (e.g., temperature-sensitive mutants).

Most ts mutants are also hypomorphic; hence we don’t see there is much advantage to their use. The addition of the AID to these proteins alone does not interfere with the ability to sustain checkpoint arrest as demonstrated in Figure S1. Instead we found that by overexpressing Rad9-AID we could demonstrate that inactivating Rad9 after 15 h behaved the same way as the inactivation of Ddc2, significantly strengthening our finding that the DDC checkpoint becomes dispensable while the SAC takes over.

(2) In cells depleted of Rad53-AID, the deletion of CHK1 stimulates an earlier release from a mitotic arrest induced by two DSBs (Figures 2D and 3C). Likewise, the authors claim that a faster escape from the cell cycle block can also be observed when upstream factors such as Ddc2, Rad9, or Rad24 are depleted in the absence of CHK1 (Figures 2A-C and Figures 3D-F). However, this earlier release from the cell cycle arrest, if at all, is only slightly noticeable in a Rad9-AID background (Figures 2B and 3E). In this sense, it is also worth pointing out that Rad9-AID chk1Δ (Figure 3E) and Rad24-AID chk1Δ (Figure 3F) cells were only evaluated up to 7 h, while in all other instances, cells were followed for 9 h, which hinders a fair assessment of the differences in the release from the cell cycle arrest.

As noted above, we have now been able to examine Rad9 over the long-time frame.

(3) Although only 25% of the cells depleted for Dun1 remained in G2/M arrest 7 h following the induction of two DSBs, it is shocking that Rad53 was nonetheless still phosphorylated after the cells had escaped the cell cycle blockage (Figure 4A).

This persistence of Rad53 phosphorylation is also seen with the inactivation of Mad2, allowing escape in spite of continued Rad53 phosphorylation.

(4) Generation of Rad9-AID2 and Rad24-AID2 strains did not fully restore the function of these proteins, since most cells had adapted 24 h after induction of two DSBs (Figure S1C). Nonetheless, Rad9-AID2 and Rad24-AID2 are still likely more stable than their AID counterparts, and hence the authors could have instead used the AID2 proteins for the experiments in Figure 2 to better evaluate the role of Rad9 and Rad24 in the maintenance of the DDC-dependent arrest.

We note again that we have found a way to study Rad9 up to 24 h.

(5) Deletion of BFA1 has been shown to promote the escape from a cell cycle arrest triggered by telomere uncapping (Wang et al. 2000, Hu et al. 2001, Valerio-Santiago et al. 2013). Likewise, while cells carrying the cdc5-T238A allele cannot adapt to a checkpoint arrest induced by one irreparable DSB, BFA1 deletion rescues the adaptation defect of this mutant CDC5 allele (Rawal et al., 2016). The authors show how, using AID-degrons of Bfa1 and Bub2, that only Bub2, but not Bfa1, is required to maintain a prolonged cell cycle arrest after the induction of two DSBs. To reinforce this point, and as shown for mad2Δ cells (Figure S6A), the authors could perform a complete time course using both the Bfa1-AID and a bfa1Δ mutant to demonstrate that they do indeed show the same behavior in terms of the adaptation to a two DSB-induced cell cycle arrest.

We thank the reviewer for noting these other instances where bfa1D promoted an escape from arrest. We tested a 2-DSB *bfa1* deletion, data has been added to Figure S9E-F. We did not observe a difference in the percentage of cells escaping arrest between the 2-DSB *bfa1* deletion and the 2-DSB *BFA1-AID* strains.

(6) Bypass or adaptation of a checkpoint-induced cell cycle arrest in *S. cerevisiae* often leads to cells entering a new cell cycle without doing cytokinesis and, hence, to the accumulation of rebudded cells. However, the experiments shown in the manuscript only account for G1 or budded cells with either one or two nuclei. Do any of the mutants show cytokinesis problems and subsequent rebudding of the cells? If so, this should have been also noted and quantified in the corresponding assays.

In the cases we have studied we have not seen instances where the cells re-bud without completing mitosis (at least as assessed by the formation of budded cells with two distinct DAPI staining masses). In the morphological assays we have done, we score the continuation of the cell cycle by the appearance of multiple buds, G1, and small budded cells. In our adaptation assays when cells escaped G2/M arrest they formed microcolonies indicating no short-term deficiency in cell division.

(7) The location of the DSB relative to the centromere of a chromosome seems to be a factor that determines the capacity of the SAC to sustain a prolonged cell cycle arrest. The authors discuss the possibility that the DSB could somehow affect the structure of the kinetochore. Did they evaluate whether Mad1 or Mad2 were more actively recruited to kinetochores in those strains that more strongly trigger the SAC after induction of the DSBs?

We have not attempted to follow Mad1/2 recruitment. ChIP-seq could be used to monitor Mad1/2 localization at the 16 centromeres in response to DSBs and the spread of g-H2AX across the centromere. Our previous data showed that g-H2AX could spread across the centromere region and could create a change that would be detected by Mad1/2. This change does not, however, affect the mitotic behavior of a strain in which the H2A genes have been modified to the possibly phosphomimetic H2A-S129E allele.

(8) The authors could speculate in the discussion about the reasons that could explain why the DDC is required for the maintenance of checkpoint arrest at early stages but then becomes dispensable for the preservation of a prolonged cell DNA DSB-induced cycle arrest, which is instead sustained at later stages by the SAC.

Our suggestion is that cells would have adapted, but modification of the centromere region engages SAC.

Finally, some minor issues are:(1) The lines in the graphs that display the results from adaptation assays (e.g., Figures 1B and 1E) or cell and nuclear morphology (e.g., Figures 1D and 1G) are too thick. This makes it sometimes difficult to distinguish the actual percentages of cells in each category, particularly in the experiments monitoring nuclear division.

Fixed

(2) While both the adaptation assay and the analysis of nuclear division in Figures 1E and 1G, respectively, show a complete DDC-dependent arrest at 4h, the Western blot in Figure 1F suggests that Rad53 is not phosphorylated at that time point. Do these figures represent independent experiments? Ideally, the analysis of cell budding and nuclear division, which is performed in liquid cultures, and the Western blot displaying Rad53 phosphorylation should correspond to the same experiment.

Cell budding in liquid cultures and adaptation assays were performed in triplicate with 3 biological replicates and the collective results are shown in each graph showing the percentage of large-budded cells. Western blot samples were collected in each liquid culture experiment. The western blot in 1G is a representative western blot.

(3) It is somewhat confusing that the blots for the proteins are not displayed in the same order in Figures 2A (Rad53 at the top) and 2B or 2C (Rad53 in the middle).

Fixed. We place Rad53 – the relevant protein - at the top.

**Reviewer #2 (Recommendations For The Authors):**
(1) Yeast with the two breaks responds to DNA damage checkpoint (DDC) until sometimes 4-15 h post DNA damage. Since the auxin-induced degradation does not completely deplete all the tagged proteins in cells, the results should be more carefully considered and not to interpret if the checkpoint entry or maintenance depends on each target protein's ability to induce Rad53 phosphorylation. It should be theoretically possible if checkpoint maintenance requires only a modest amount of checkpoint factors especially because the experiments involve the induction of one or two DSBs. The low levels of DDC factors may be insufficient for Rad53 activation but could still be effective for cell cycle arrest. Indeed, the Haber group showed that the mating type switch did not induce Rad53 phosphorylation but still invoked detectable DNA damage response. To test such possibilities, the authors might consider employing yet another marker for DDC such as H2A or Chk1 phosphorylation besides Rad53 autophosphorylation. Alternatively, the authors might check if auxin-induced depletion also disrupts break-induced foci formation for checkpoint maintenance or their enrichment at DNA breaks using ChIP assays at various points post-damage.

DAPI staining of Ddc2-AID cells show that when IAA is added 4 h after DSB induction (Figure S3A), cells escape G2/M arrest as evidenced by the increase in large-budded cells with 2 DAPI signals, small budded cells, and G1 cells. Overexpression of Ddc2 can sustain the checkpoint past 24 h, but without SAC proteins like Mad2 they will eventually adapt (Figure S6B).

That Rad9-AID or Rad24-AID in the absence of added auxin (but in the presence of TIR1) is unable to sustain arrest suggests to us that low levels of Rad9 or Rad24 are not sufficient to maintain arrest. As the reviewer notes, normal MAT switching doesn’t cause Rad53 phosphorylation or arrest, though early damage-induced events such as H2A phosphorylation do occur. But our point is that Rad9 or Ddc2 is needed to maintain arrest only up to a certain point, after which they become superfluous and a different checkpoint arrest is imposed. At that point apparently a low level of these proteins plays no obvious role.

(2) It is interesting that DDC no longer responds to the damage signaling after 15 h of DSB-induced prolonged checkpoint arrest after two DNA double-strand breaks. Is this also applicable to other adaptation mutants? The results might improve the broad impact of the current conclusions. It is also possible that the transition from DDC to SPC depends on simply the changes in signaling or in part due to the molecular changes in the status of DNA breaks or its flanking regions. Indeed, the proposed model suggests that the spreading of H2A phosphorylation to centromeric regions induces SAC and thus mitotic arrest. The authors could measure H2A phosphorylation near the centromere using ChIP assays at various intervals post-DNA damage. It is particularly interesting if depletion of Ddc2 at 15 h post DNA damage does not alter the level of H2A phosphorylation at or near centromere.

Our previous data have suggested that the involvement of the SAC in prolonging DSB-induced arrest involved post-translational modification of centromeric chromatin such as the Mec1- and Tel1-dependent phosphorylation of the histone H2A (Dotiwala). In budding yeast there is also a similar DSB-induced modification of histone H2B (Lee et al.). To ask if there is an intrinsic activation of the SAC if the regions around centromeres were modified by checkpoint kinase phosphorylation, we examined cell cycle progression in strains in which histone H2A or histone H2B was mutated to their putative phosphomimetic forms (H2A-S129E and H2B-T129E). As shown in Figure S11, there was no effect on the growth rate of these strains, or of the double mutant, suggesting that cells did not experience a delay in entering mitosis because of these modifications. We note that although histone H2A-S129E is recognized by an antibody specific for the phosphorylation of histone H2A-S129, the mutation to S129E may not be fully phosphomimetic.

(3) It is puzzling why Rad9-AID or Rad24-AID are proficient for DDC establishment but cannot sustain permanent arrest in the two break cells. It appears Rad53 phosphorylation for DDC is weaker in cells expressing Rad9-AID or Rad24-AID according to Fig.2B and C even though their protein level before IAA treatment is still robust. This might also explain why the results of depleting Rad53 and Rad9 are very different. It also raises concern if the effect of Rad24 depletion on checkpoint maintenance is in part due to the weaker checkpoint establishment. It might be necessary to use the AID2 system to redo Rad24 depletion to exclude such a possibility.

We believe that the AID mutants are very sensitive to the low level of IAA present in yeast. The instability of the protein is entirely dependent on the TIR1 SCF factor, so the proteins themselves are not intrinsically defective; they are just subject to degradation. Overexpressing Rad9 allowed us to evaluate its role at late time points.

(4) It is intriguing that the switch from DDC to SAC might take place at around 12 h when yeasts with a single unrepairable break ignore DDC and resume cell cycling (so-called "adaptation"). Since 4h and 15h are far apart and the transition point from DDC to SAC likely takes place between these two points, it will be very helpful to analyze and compare cell cycle exit after 24 h by treating IAA at multiple points between 4-15h.

When we add IAA to Mad2-AID and Mad1-AID 4 h after DSB induction, cells remain arrested for up to 12 h after DSB induction. At 15 h cells begin to exit checkpoint arrest indicating that the handoff of checkpoint arrest must occur between 12 to 15 h after DSB induction. If we degraded DNA damage checkpoint proteins at any point before Mad2, Mad1, and Bub2 begin to contribute to checkpoint arrest, then arrested cells will likely adapt in a similar manner to when IAA was added 4 h after DSB induction.

(5) Some of the Western blot quality is poor. For instance, in Figure 6C, Mad1-AID level after IAA addition is not compelling especially because the TIR level (the loading control) is also very low.

In Figure 6C, while the relative levels of TIR1 are similar in the IAA treated and untreated samples, there is no detectable amount of Mad1-AID in the IAA treated samples indicating that Mad1-AID was successful degraded with the AID system.

(6) Fig. 8 is complex. It might be helpful to define the different types of arrows in the figure. The legend also has a spelling error, Rad23 should be Rad24.

We’ve defined what each arrow means in the legend and corrected the spelling error in the figure legend.

**Reviewer #3 (Recommendations For The Authors):**
Major concerns:Much of the manuscript states that two unrepairable DSBs lead to a long and severe G2/M arrest. Two main cytological approaches are used to make this statement: bud size and number on plates after micromanipulation (microcolony assay), and cell and nuclear morphology in liquid cultures. While the latter gives a clear pattern that can be assigned to a G2/M block as expected by DDC, i.e. metaphase-like mononucleated cells with large buds, the former can only tell whether cells eventually reach a second S phase (large budded cells on the plate can be in a proper G2/M arrest, but can also be in an anaphase block or even in the ensuing G1). The authors always performed the microcolony assay, but there are several cases where the much more informative budding/DAPI assay is missing. These include Dun1-aid and others, but more importantly chk1D and its combinations with DDC proteins. Incidentally, for the microcolony assay, it is more accurate to label the y-axis of the corresponding graphs (and in the figure legends and main text) with something like "large budded cells"; "G2/M arrested cells" is misleading.

Figures have been updated to more accurately reflect what we are measuring.

The results obtained with the Bfa1/Bub2 partner are intriguing. These two proteins form a complex whose canonical function is to prevent exit from mitosis until the spindle is properly aligned, acting in a distinct subpathway within the SAC that blocks MEN rather than anaphase onset. The data presented by the authors suggest that, on the one hand, both SAC subpathways work together to block the cell cycle. However, why does canonical SAC (Mad1/Mad2) inactivation not lead to a transition from G2/M (metaphase-like) arrested cells to anaphase-like arrest maintained by Bfa1-Bub2? Since Bfa1-Bub2 is a target of DDC, is it possible that DDC knockdown also inactivates this checkpoint, allowing adaptation? On the other hand, can the authors provide more data to confirm and strengthen their claim of a Bfa1-independent Bub2 role in prolonged arrest? Perhaps long-term protein localization and PTM changes. Bub2-independent roles for Bfa1 have been reported, but not vice versa, to the best of my knowledge.

In the mitotic exit network Bfa1/Bub2 prime activation of the pathway by bringing Tem1 to spindle pole bodies. Phosphorylation of Bfa1 causes Tem1 to be released and phosphorylate Cdc5 to trigger exit by MEN. It has been shown that DNA damage, in a *cdc13-1* ts mutant, phosphorylates Bfa1 in a Rad53 and Dun1 dependent manner. This phosphorylation of Bfa1 could release Tem1 and prime cells to exit checkpoint arrest when cells pass through anaphase. Looking at Tem1 localization to spindle pole bodies and interactions with Bfa1/Bub2 in response to DNA damage might give insight into why cells don’t experience an anaphase-like arrest when they are released by either deactivation of the DNA damage checkpoint or SAC.

We have previously shown that a deletion of *bub2* in a 1-DSB background shortens DSB-induced checkpoint arrest. Deletion of *bfa1* in a 2-DSB background showed ~80-70% of cells stuck in a large-budded state as measured through an adaptation assay tracking the morphology of G1 cells on a YP-Gal plate and DAPI staining. Deletion or degradation of *bfa1* might not release cells from arrest because the Mad2/Mad1 prevent cells from transitioning into anaphase. Our DAPI data for Bub2-AID shows an increase in cells with 2 DAPI signals (transition into anaphase) and small budded cells indicating that degradation of Bub2 is releasing cells into anaphase and allowing cells to complete mitosis.

Further suggestions:It would be richer if authors could provide more than one experimental replicate in some panels (e.g., S1A,B; S4A; and S6B).

S1C confirms that Rad9-AID and Rad24-AID will adapt by 24 h even with the point mutant TIR1(F74G) which has lower basal degradation than TIR1. S4A has been updated with additional experimental replicates. The 48 h timepoint after DSB induction was to show the importance of Mad2 even when Ddc2 is overexpressed.

Figure 1: Rearrange figure panels when they are first mentioned in the text. For example, it makes more sense to have the plate adaptation assay as panel B for both 1-DSB and 2-DSB strains, budding plus DAPI as panel C, and Rad53 as panel D.

These figures have been rearranged in the order that they are mentioned in the paper.

Figure 5: Correct Ph-5-IAA in the Rad53 WBs (it should be 5-Ph-IAA).

This has been corrected.

Figure S2: The straight line under the "+IAA" text box is misleading. I think it should also cover the "-2" time point, right? Also, check the figure legend. Information is missing and does not correspond to the figure layout.

This has been corrected.

Figure S3: Perhaps "Cell cycle profile as determined by budding and DAPI staining" is a better and more accurate legend title.

The legend title has been updated to “Cell cycle profile as determined by budding and DAPI staining in Ddc2-AID and Rad53-AID mutants ± IAA 4 h after galactose.”

Figure S5: Detection of both Rad53 and Ddc2 in the same blot could lead to misinterpretation as hyperphosphorylated Rad53 appears to coincide with Ddc2 migration.

Figure S5A-B are representative western blots where Rad53 was probed to show activation of the DNA damage checkpoint by Rad53 phosphorylation. When measuring the relative abundance of Ddc2 we did not probe all blots for Rad53.

Table S1: Include the post-hoc test used for comparisons after ANOVA.

A Sidak post-hoc test was used in PRISM for the one-way ANOVA test. PRISM listed the Sidak post-hoc test as the recommended test to correct for multiple comparisons. A column has been added to S. Table 1 to show which post-hoc test was used.

Page 10, line 4: The putative additive effect of chk1 knockout with Dun1 depletion should also be compared to chk1 alone (in Figure 3A).

We address the additive effect of *chk1* knockout with Dun1-AID depletion in a later section on Page 11, line 6. Since we had not explored possible effects from downstream targets of Rad53 for prolonging checkpoint arrest when Rad53 was depleted, we did not mention the effect of the *chk1* knockout on Dun1 depletion.

Page 14, second paragraph, line 4: "Figure 6A-D", is it not?

Figure S6A is measuring checkpoint arrest in a deletion of *mad2* in a 2-DSB strain. Figure 6A-D shows how degradation of Mad2-AID and Mad1-AID after the handoff of arrest causes cells to exit the checkpoint in a Rad53 independent manner.